# Nucleosome plasticity is a critical element of chromatin liquid–liquid phase separation and multivalent nucleosome interactions

Stephen E. Farr[1], Esmae J. Woods [1], Jerelle A. Joseph [1,2,3], Adiran Garaizar [1] &
Rosana Collepardo-Guevara [1,2,3 ✉]

Liquid–liquid phase separation (LLPS) is an important mechanism that helps explain the membraneless compartmentalization of the nucleus. Because chromatin compaction and LLPS are collective phenomena, linking their modulation to the physicochemical features of nucleosomes is challenging. Here, we develop an advanced multiscale chromatin model—integrating atomistic representations, a chemically-specific coarse-grained model, and a minimal model—to resolve individual nucleosomes within sub-Mb chromatin domains and phase-separated systems. To overcome the difficulty of sampling chromatin at high resolution, we devise a transferable enhanced-sampling Debye-length replica-exchange molecular dynamics approach. We find that nucleosome thermal fluctuations become significant at physiological salt concentrations and destabilize the 30-nm fiber. Our simulations show that nucleosome breathing favors stochastic folding of chromatin and promotes LLPS by simultaneously boosting the transient nature and heterogeneity of nucleosome–nucleosome contacts, and the effective nucleosome valency. Our work puts forward the intrinsic plasticity of nucleosomes as a key element in the liquid-like behavior of nucleosomes within chromatin, and the regulation of chromatin LLPS.

[1] Maxwell Centre, Cavendish Laboratory, Department of Physics, University of Cambridge, Cambridge, UK. [2] Yusuf Hamied Department of Chemistry, University of Cambridge, Cambridge, UK. [3] Department of Genetics, University of Cambridge, Cambridge, UK. ✉email: rc597@cam.ac.uk

The Eukaryotic nucleus is a highly compartmentalized system that achieves its internal organization entirely without the use of membranes[1]. Inside the nucleus, hundreds of millions of DNA base pairs are densely packed into a highly dynamic and heterogeneous structure known as chromatin[2]. The basic building blocks of chromatin are nucleosomes: 10-nm wide nanoparticles composed of approximately 147 base pairs (bp) of DNA wrapped around a histone protein octamer (two copies each of H2A, H2B, H3, and H4)[3,4]. To assemble chromatin, nucleosomes are first joined together by free DNA linker segments of varying lengths—measured in units of nucleosome repeat lengths (NRL = 147 bp + linker DNA length)— forming a "beads-on-a-string" structure termed the 10-nm fiber[5]. Subsequently, using their charged and contoured surfaces[6], and charged and flexible protruding "arms" (histone tails), nucleosomes establish interactions with one another and with the DNA[7]. These interactions trigger folding of the 10-nm fiber and the dense packing of DNA inside cells[8,9].

The structure of chromatin, beyond the 10-nm fiber, remains an intense topic of research and debate[2,10–12]. The traditional textbook view suggests that 10-nm chromatin folds into a regular and rigid 30-nm zigzag fiber, where nucleosomes interact preferentially with their second-nearest neighbors (i.e., nucleosome $i$ with $i \pm 2$). While this zigzag fiber model is supported by in vitro studies of reconstituted chromatin arrays[13–16], experiments interrogating chromatin inside cells have consistently failed to detect 30-nm fibers[17–20]. Instead, accumulating evidence is shifting the structural paradigm of chromatin in vivo in favor of the "liquid-like" or "fluid-like" model[2,19], which proposes that 10-nm fibers condense into an irregular and dynamic polymorphic ensemble[10,18,21]. The term "liquid" here is used to emphasize a structure that is absent of long-range translational order, where nucleosomes can flow and relax easily, and engage in interactions with a wide range of neighbors[2,18,19], like in a "sea of nucleosomes"[22]. Nucleosomes within disordered chromatin are proposed to form heterogeneous groups of variable sizes and densities—interspersed with nucleosome-free regions[23]. Furthermore, the density of nucleosomes, rather than the structure of the 30-nm fiber, is what seems to distinguish different chromatin regions (e.g., nucleosome density is higher in heterochromatin than in euchromatin)[16,23], and change upon differentiation[24]. Sequencing-based methods that can resolve chromatin interactions in situ at single-nucleosome resolution—i.e., micrococcal nuclease chromosome conformation assay (Micro-C)[25,26], ionizing radiation-induced spatially correlated cleavage of DNA with sequencing (RICC-seq)[27], and high-throughput chromosome conformation capture with nucleosome orientation (Hi-CO)[28]— suggest that the irregular organization of chromatin is underpinned by dominant interactions among $i$ and $i \pm 2$ nucleosomes. Consistent models where chromatin exhibits a large-scale disordered organization but contains strong short-range zigzag contacts include the hierarchical looping model[29,30] and the multiplex higher-order folding model[31].

The disordered behavior of nucleosomes is not surprising if one considers the notable intrinsic heterogeneity of chromatin physicochemical parameters in vivo (e.g., varying DNA sequences and epigenetic marks, heterogeneous distributions of post-translational histone modifications, non-uniform NRLs, presence of nucleosome-free regions, and dynamic nucleosome breathing and sliding motions). Many of these parameters can independently enable chromatin polymorphism, triggering the folding of 10-nm fibers into irregular loops, hairpins, and bends[21]. Indeed, irregular nucleosome spacing[21,32], nucleosome-free regions[23,24,32–34], heterogeneous on/off dyad binding of linker histone proteins to the nucleosome[35,36], low linker histone concentrations or subtype variations[36], inhomogeneous

distributions of post-translational modifications[37], and the disordered nature of the linker histone protein[38] can independently give rise to a plethora of nucleosome orientations and interactions. In concert, these factors can further amplify or control chromatin polymorphism[34,37,39].

In the past three years, the paradigm of dynamic liquid-like behavior of nucleosomes within cells has gained significant traction due to the realization that chromatin and its associated multivalent biomolecules can undergo liquid–liquid phase separation (LLPS) in vitro and in cells[40–52]. LLPS is now postulated as a mechanism, alongside others[53], to explain genome compartmentalization without the use of physical membranes. Notable examples include the formation of constitutive heterochromatin via the transcriptional repressor HP1[40,41,52], as well as the emergence of important liquid-like structures such as the nucleoli, Cajal bodies, and nuclear speckles[54]. In addition, the formation of super-enhancer regions has been recently associated with the phase separation of a combination of transcription factors and co-factors, chromatin regulators, non-coding RNAs, and RNA Polymerase II[44–46,49–51].

The origin of intranuclear phase separation is intricately linked to the complex and crowded biomolecular environment of the cell nucleus[55,56]; i.e., the nucleoplasm is a highly multi-component mixture of proteins and nucleic acids with varying compositions across different regions[54]. Indeed, orientation-independent differential interference contrast microscopy reveals that the density of the total material in the nucleus is as high as 208 mg/ml within heterochromatin and 136 mg/ml in the surrounding euchromatin regions[57]. The formation of diverse phase-separated chromatin compartments becomes thermodynamically stable in specific genomic regions when the concentration of key biomolecules, termed scaffolds[58], surpasses a threshold. These conditions allow scaffolds—normally dissolved in the nucleoplasm—to drive LLPS by minimizing their free energy through the formation of numerous attractive interactions with one another. Therefore, the features that affect binding among nucleosomes, and between nucleosomes and their chromatin-binding proteins (e.g., their chemical makeup and mechanical properties, along with the prevailing microenvironment) are expected to be crucial regulators of intranuclear LLPS. In particular, the capacity of biomolecules (e.g., proteins, RNA, DNA, and nucleosomes) to interact with at least three binding partners (i.e., multivalency) is essential for forming sufficient transient interconnections to compensate for the entropic loss due to the reduced number of microstates upon demixing[59,60].

Our work focuses on assessing an important feature of nucleosomes that likely pertains to chromatin LLPS: their inherent plasticity. That is, rather than static building blocks—as routinely considered in large-scale chromatin structural models— at short length-scales nucleosomes are highly dynamic and structurally irregular entities[4,61–66], and are better described as a "dynamic family of particles"[67]. Nucleosomes in vivo can pack a broad range of DNA base pairs around the histone core (~100–170 bp), and can also have varied histone compositions and stoichiometries[67]. In addition, thermal fluctuations cause some of the DNA–histone core interactions to spontaneously break and reform from one end of the nucleosome, while the majority of nucleosomal DNA remains wrapped around the histone core[4,64]—these phenomena, known as "nucleosome breathing" (also referred to as DNA breathing), is favored at physiological salt concentrations[68,69]. In vivo, the probability of nucleosome breathing across chromatin is likely very diverse too, as it can be sensibly altered by post-translational modifications of nucleosomes[70–72] and based on their constituent DNA sequences[71,73,74]. Furthermore, the presence of H3K56Ac combined with DNA sequence changes can increase the rate of

nucleosome unwrapping by one order of magnitude or more[71]. The recruitment of chromatin-binding proteins can also affect the plasticity of nucleosomes. For instance, binding of multiple Swi6 molecules to a nucleosome has been shown to disrupt the DNA–histone bound state, increasing the exposure of the buried histone core to solvent, and crucially, promoting chromatin LLPS[52].

Nucleosome structural fluctuations provide a transient opportunity for the binding of transcription factors to DNA, and hence, for transcription[75,76]. However, prior to the nucleosome barrier, pioneer and other transcription factors need to surpass the steric hindrance imposed by nucleosome–nucleosome interactions. This seems particularly challenging if we consider the apparent rigidity of nucleosomes within the 30-nm fiber structural models. Therefore, this begs a fundamental question that we target in this work: What are the physical and molecular factors that modulate the accessibility of nucleosomes within compact assemblies? To resolve this conundrum, assessing how the mesoscale properties of chromatin (e.g., density, flexibility, shape, and size) are impacted by the dynamic behavior of nucleosomes is crucial. In this regard, computational modeling of chromatin structure—both top-down polymer models trained on experimental datasets[77,78] and bottom-up mechanistic descriptions[79]—offers an ideal complement to experiments. Polymer chromatin models trained on experimental datasets[77,78] are currently the best tools for investigating chromatin organization at the whole-nucleus scale; these approaches are essential to propose plausible structural ensembles of chromatin and test whether hypotheses on the structural behavior of chromatin and/or the physical mechanisms that dictate such structure are consistent with the experimental data. In partnership, mechanistic computational models[79] can generate hypotheses and link them to the physicochemical properties of chromatin. A wide range of mechanistic coarse-grained models with nucleosome and sub-nucleosome resolution have been developed in the past few years[8,9,15,21,29,30,80–97] to bridge molecular and physicochemical information of nucleosomes to the mesoscale properties of chromatin. Future integration of both data-driven polymer models with mechanistic chromatin descriptions holds great potential for providing a complete view of chromatin organization.

In the present study, we develop an advanced mechanistic multiscale chromatin model designed to investigate the connection between the fine molecular details of nucleosomes (including amino acid and DNA sequence, specific distributions of post-translational modifications and epigenetic marks, protein flexibility, DNA mechanical properties, and nucleosome plasticity) and the mesoscale (up to sub-Mb scale) organization of chromatin. By bridging atomistic and sub-Mb chromatin scales, our multiscale approach can dissect biophysical properties of chromatin that emerge from the dynamic formation and breakage of interactions among a few thousand nucleosomes with diverse molecular features; thereby, it can uncover the molecular and biophysical mechanisms that explain the self-organization and intrinsic LLPS of chromatin with diverse chemical makeups. When we model chromatin at residue/base-pair resolution, the dominant role of electrostatics, the high dimensionality of the system, and the resemblance of chromatin to a poly-branched polymer make attaining sufficient sampling highly non-trivial. To overcome these challenges, we develop, in tandem, a powerful Debye-length replica-exchange molecular dynamics (MD) simulation approach that is transferable and can be used to explore the energy landscapes of other intractable charged systems.

Our multiscale model and Debye-length enhanced-sampling technique reveal that nucleosome breathing is promoted at physiological salt conditions, in agreement with experiments[68,69].

Whereas enhancement of nucleosome breathing at physiological salt concentrations drives chromatin to populate a highly dynamical, but compact, liquid-like structural ensemble[2,19], inhibition of breathing at low salt gives rise to 30-nm zigzag fibers. This modulation of nucleosome breathing with salt, and its impact on chromatin self-assembly might help reconcile longstanding differences between fiber-based and in vivo chromatin models. Our simulations further explain that liquid-like chromatin organization is characterized by short-lived and orientationally diverse internucleosome interactions, which are mediated by transient non-specific DNA–histone tail contacts. In contrast, 30-nm fibers are sustained by long-lived regular face-to-face nucleosome interactions. Importantly, nucleosome plasticity promotes both liquid-like folding of individual chromatin systems and LLPS of chromatin arrays via the same physical mechanism: namely, it enhances the multivalency of nucleosomes and, therefore, the connectivity and stability of both compact chromatin and the phase-separated chromatin. The stochastic organization of nucleosomes within compact chromatin that we observe, both within single arrays and condensates, paints a much more permissive picture of nucleosome targeting than that offered by the fiber-based models. The realization that nucleosomes can be simultaneously stochastically organized and tightly packed might have important implications in reshaping the molecular mechanisms used to link chromatin structure to modulation of DNA accessibility.

## Results

**Multiscale model for heterogeneous chromatin.** We have developed a multiscale model (Fig. 1 and Supplementary Fig. 1) that describes the physicochemical heterogeneity of chromatin and the plasticity of in vivo nucleosomes, as well as their impact on functionally relevant length scales (i.e., up to sub-Mb scales). This model integrates three complementary levels of resolution: atomistic representations of nucleosomes (Level 1), a chemically-specific chromatin model (Level 2), and a minimal chromatin model (Level 3). Integration of these spatiotemporal scales is necessary to probe the biophysical and molecular forces underpinning the modulation of large-scale chromatin organization by subtle chemical changes (e.g., in charge, hydrophobicity and flexibility) originating, for instance, from histone/DNA mutations, post-translational modifications of histones, DNA epigenetic marks, and binding of regulatory proteins.

Our chemically-specific coarse-grained model features representations of breathing nucleosomes with all histone proteins resolved at the residue level (preserving the sequence-dependent charge, hydrophobicity, size, and flexibility of the atomistic histones). Double-stranded DNA is described at the base-pair step level (with charge and sequence-dependent mechanical properties described with a modified version of the Rigid Base Pair (RBP)[98–102] model with added phosphate charges; see "Methods" section). Parameters for our chemically-specific model are obtained from experimental amino-acid pairwise contact propensities[103–105], large datasets of atomistic MD simulations of DNA strands[106], and bias-exchange metadynamics atomistic simulations of 211-bp nucleosomes[38].

Notably, the physicochemical and molecular fidelity of histones and DNA within our chemically-specific coarse-grained model gives rise to a DNA polymer that spontaneously wraps ~1.7 times around the histone core, and that adopts the correct topology and left-handed chirality under weak negative DNA supercoiling, and the chiral inverted right-handed counterpart under weak positive supercoiling, consistent with experiments[107–110] (Supplementary Notes and Supplementary Fig. 8). The molecular resolution of our model also results in nucleosomes that naturally exhibit

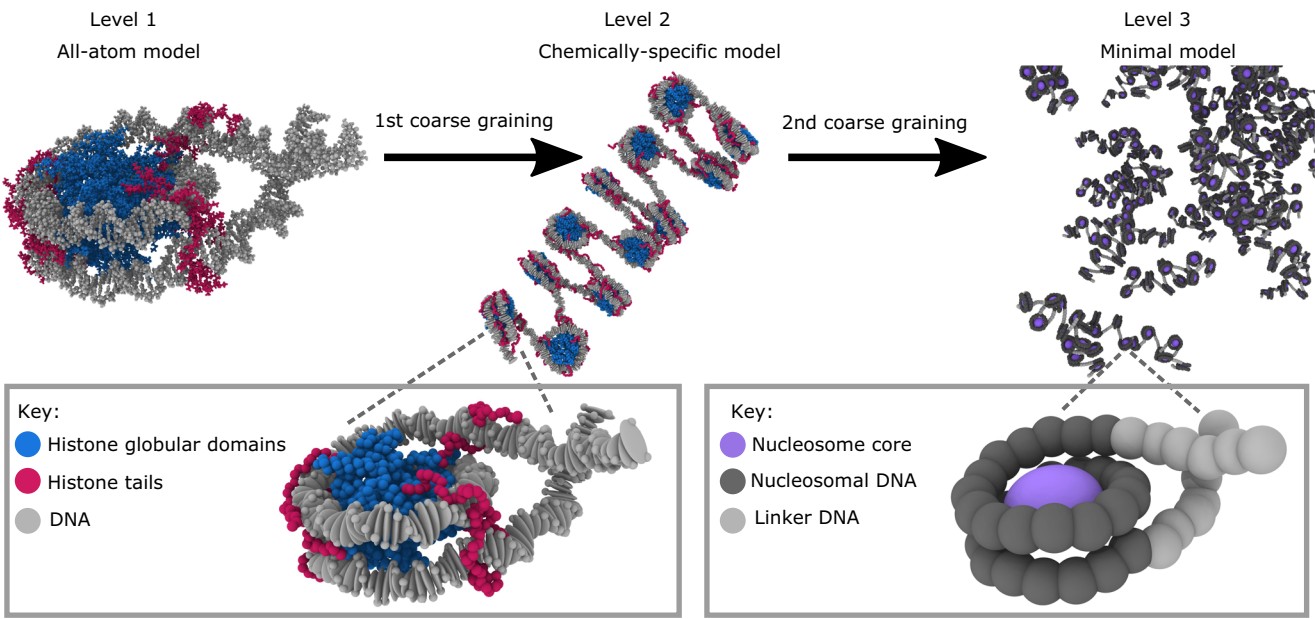

**Fig. 1 Schematic representation of our chromatin multiscale toolkit spanning three levels of resolution.** (Level 1) Our high-resolution layer: Atomistic MD simulations of DNA and nucleosomes to identify key physicochemical information. An atomistic nucleosome is depicted with atoms in the histone globular domains colored in blue, in the histone tails in magenta, and in the DNA in gray. (Level 2) Our mid-resolution layer: Debye-length replica-exchange MD simulations of our chemically-specific coarse-grained chromatin model; representing DNA at the base-pair level (one gray ellipsoid per base-pair) and modeling histone tails (one magenta bead per amino acid) and histone globular domains (one blue bead per amino acid) at the residue level. This model links elementary properties of nucleosomes to mesoscale behavior of oligonucleosomes by accounting for the following key molecular features from Level 1: DNA mechanical properties, secondary structure of the histone globular regions, flexibility of histone tails, and the size, shape, electrostatics, and hydrophobicity of individual amino acids and base pairs. (Level 3) Our low-resolution layer: Direct coexistence simulations of our minimal coarse-grained chromatin model designed to investigate the phase behavior of a few thousand interacting nucleosomes. The histone core is treated as a single interaction site (purple bead) with parameters from internucleosome potential of mean force simulations from Level 2. The nucleosomal DNA (dark gray beads) and the linker DNA (light gray beads) are described explicitly (1 bead = 5 bp) with our minimal RBP-like model parameterized from free DNA simulations from Level 2.

spontaneous breathing motions (i.e., without being primed to do so) and display force-induced unwrapping in quantitative agreement with experiments at single base-pair resolution (see "Model validation" section, Supplementary Notes, and Fig. 2).

Further coarse-graining is essential to reduce the system dimensionality and investigate chromatin LLPS. Accordingly, from our chemically-specific coarse-grained simulations, we derive a consistent minimal chromatin model that describes each nucleosome with just a few particles. Such a dimensionality reduction enables the simulation of chemically heterogeneous sub-Mb scale chromatin regions and LLPS, while also considering nucleosome thermal fluctuations. Specifically, we use one ellipsoid for each histone core, and develop a "minimal RBP-like model" for DNA at a resolution of 5 bp per bead, with minimal helical parameters extracted from chemically-specific coarse-grained simulations of 200 bp DNA strands (see "Methods" section and the Supplementary Methods). Nucleosome–nucleosome and nucleosome–DNA interactions are modeled with orientationally dependent potentials fitted to reproduce internucleosome potentials of mean force calculated with our chemically-specific coarse-grained chromatin model (Supplementary Fig. 4 and Supplementary Methods). Significantly, a comparison of the rate of exponential decay of the autocorrelation functions of the chromatin radius of gyration in both models (Supplementary Fig. 7 and Supplementary Notes) indicates that the timescales in our minimal chromatin model are 10 times faster than those in our chemically-specific model.

The detailed description of our models and the rationale behind their designs, resolutions, and parameters are given in the "Methods" section and the Supplementary Methods. A comparison of our model predictions with experiments is discussed in the "Model validation" section below and in the Supplementary Notes.

**Model validation**. We begin by comparing various quantities in our chemically-specific coarse-grained chromatin model with corresponding experimental observables. First, we find that the persistence length of DNA estimated in our simulations (i.e., performed using our modified RBP model with phosphate charges; see "Methods" section) agrees well with salt-dependent values from high-throughput tethered particle motion single-molecule[111] and light scattering[112] experiments, and with sequence-dependent measurements from cyclization assays[113] (Supplementary Fig. 9 and Supplementary Notes). The residue-resolution protein model[103] that we use to describe histones has been shown to reproduce well the experimental radii of gyration of several intrinsically disordered proteins[105].

Next, we assess the accuracy of the critical dynamical unwrapping behavior of nucleosomes in our model by performing single-molecule force-extension simulations and comparing our results with those from force spectroscopy experiments[114–117]. Pulling a single nucleosome at the equilibrium speeds typically used in force spectroscopy experiments (e.g., ~0.1 mm/s for magnetic tweezers[117]) is computationally unfeasible; that would require millisecond-long trajectories, which are presently not easily achievable. With the current computing power, force-extension steered MD would need to be performed at pulling speeds significantly above those required to maintain equilibrium conditions (e.g., 10–100 times faster than in experiments). To overcome this computational limitation,

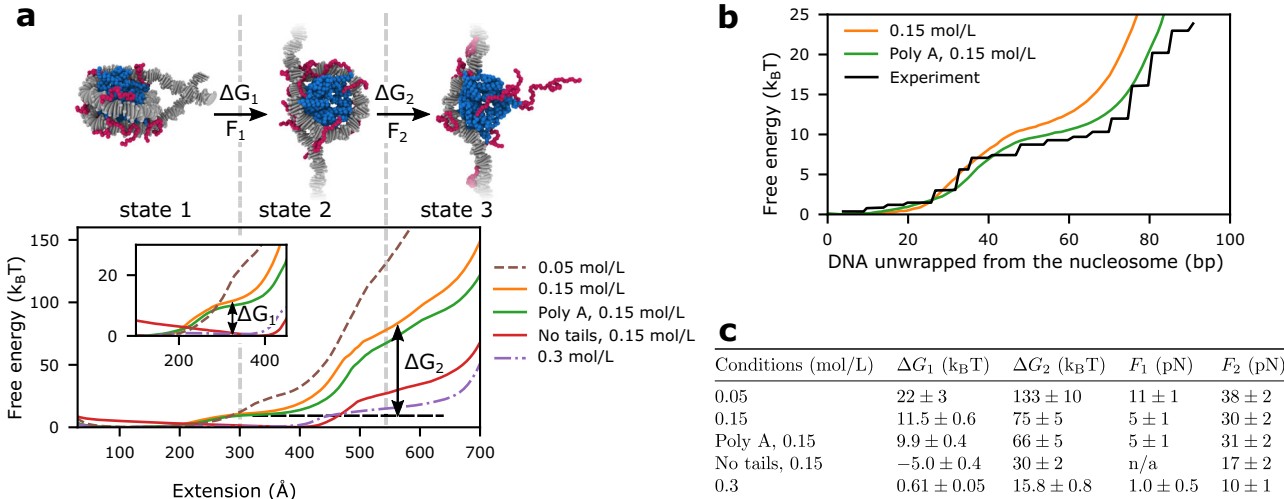

**Fig. 2 Validation of chemically-specific coarse-grained model against force spectroscopy experiments. a** Model predictions for the force-induced unwrapping of nucleosomes under varying conditions. Top: Representative simulation snapshots of nucleosome configurations (color-coded as in Fig. 1—level 2) at three different stages of the unwrapping process, showing a fully wrapped nucleosome (state 1) at low pulling forces ($\leq F_1$ in Table in **c**), a nucleosome with the first turn unwrapped (state 2) at intermediate forces ($F_1$–$F_2$ in Table in **c**), and a fully unwrapped nucleosome (state 3) at higher forces ($\geq F_2$ in Table in **c**). $F_1$ is the maximum force during the state 1 to state 2 transition, $F_2$ is the maximum force during the state 2 to state 3 transition. Bottom: Free-energy cost in units of $k_BT$ for nucleosome unwrapping as measured by the PMF as a function of the end-to-end DNA distance (or extension) measured in Angstroms (Å). The dashed brown, solid orange, and dashed purple curves correspond to 1KX5-nucleosome[7] simulations at 0.05, 0.15, and 0.3 mol/L of NaCl, respectively. The green curve corresponds to simulations of a polyA nucleosome at 0.15 mol/L NaCl. The red curve was calculated for a nucleosome with all histone tails clipped at 0.15 mol/L NaCl. The inset provides a zoomed-in view of the low-force regime and indicates the free-energy difference between states 1 and 2 ($\Delta G_1$) for the simulations of 1KX5 nucleosomes at 0.15 mol/L NaCl. The vertical dashed lines are used as visual aids to guide the reader to the approximate regions on the PMF that exhibit different states of nucleosome unwrapping. The horizontal dashed line highlights the free-energy plateau corresponding to the transition between states 1 and 2 of the 1KX5 nucleosomes at 0.15 mol/L NaCl; the free-energy difference between states 2 and 3 ($\Delta G_2$) is also illustrated for this case. $\Delta G_1$ is the free-energy difference between states 1 and 2. $\Delta G_2$ is the free-energy difference between states 2 and 3. **b** Quantitative agreement between the free-energy cost of nucleosome unwrapping at single DNA base-pair resolution estimated with our simulations at 0.15 mol/L NaCl for nucleosomes with the 1KX5 sequence (orange curve) and a polyA sequence (green curve), and that derived from analysis of mechanical unzipping experiments at 0.10 mol/L NaCl and 0.5 mmol/L MgCl₂ (black curve)[119,120]. **c** Summary of the change in free energy (mean ± standard deviation) between nucleosome unwrapping states, and the corresponding rupture forces. n/a for no tails denotes that there is no rupture force for state 1 to state 2 transition as the free energy minimum is state 2. The values of $G$ and $F$ are obtained from reading off the graphs in **a**. and the uncertainty values come from the range of positions that these values could be read from.

following the procedure of Lequieu et al.[118], we perform equilibrium umbrella sampling to estimate the potential of mean force (PMF) of single nucleosome unwrapping, using the DNA end-to-end distance as the order parameter. Subsequently, we derive force-extension plots by taking the numerical derivative of the PMF curves (see force-extension curves in Supplementary Fig. 10).

Consistent with experiments, the force-induced nucleosome unwrapping behavior predicted by our model can be separated into three equilibrium regimes (Fig. 2a, b), each spanning a force-extension region (Supplementary Fig. 10) that matches the experimental values in refs. [114–117] (see further discussion in Supplementary Notes). Most notably, by modeling nucleosomes with chemical and mechanical accuracy, we obtain quantitative agreement between our model predictions and the free-energy landscape for nucleosome unwrapping[119] derived from mechanical unzipping experiments at single DNA base-pair resolution[120] (Fig. 2c).

Further, our model shows that unwrapping of the outer DNA turn is associated with a free-energy barrier ($\Delta G_1$ in Fig. 2) of ~11.5 $k_BT$ at 0.15 mol/L NaCl, which closely matches estimates from force spectroscopy experiments at similar salt conditions (~9–11.1 $k_BT$)[115,117,121]. In agreement with magnetic tweezer experiments[121], we estimate an increase of ~10.5 $k_BT$ in the free-energy barrier for outer DNA turn unwrapping as the monovalent salt concentration decreases from 0.15 to 0.05 mol/L of NaCl (Fig. 2a, c). This increase in free energy can be explained

from the enhanced electrostatic attraction between DNA and histones at low salt. Hence, as previously shown, nucleosome breathing is feasible at physiological salt concentrations but becomes increasingly challenging in low salt conditions[68].

The free-energy landscape for nucleosome unwrapping computed with our model agrees well with several other experimental findings. For instance, in line with the dependency of nucleosome unwrapping on DNA sequence[71], nucleosomes with an unfavorable polyA DNA sequence have a free-energy barrier for unwrapping the outer DNA turn that is 1.5 $k_BT$ smaller than in 1KX5 nucleosomes[7]. Histone tail clipping leads to the unwrapping of the outer turn with a negligible energetic penalty, in agreement with mechanical disruption experiments where unwrapping of the outer DNA turn occurs at near-zero forces after histone tail removal[114].

Regarding chromatin behavior, our force-extension curves computed using an extension clamp (Supplementary Fig. 10) exhibit the typical saw-toothed pattern of optical tweezer experimental curves[122], where the force displays an abrupt drop accompanied by an increase in the extension due to the partial unwrapping of individual nucleosomes (see further discussion in the Supplementary Notes). Such behavior is also consistent with the step-like patterns emerging from magnetic tweezer experiments[117], where a force clamp is used instead. As we describe below, our sedimentation coefficients for 12-nucleosome 165-bp chromatin arrays are in quantitative agreement with the experimental values of Grigoryev and colleagues[123]. In addition,

the liquid-like chromatin ensembles that we report are in qualitative agreement with the disordered organization of nucleosomes within clutches derived from super-resolution microscopy experiments of chromatin in nuclei[23], and the chromEMT polymorphic nucleosome organization within chromatin in situ[16].

**Nucleosome plasticity underlies the stochastic folding of chromatin.** Our chemically-specific coarse-grained chromatin model contains sufficient physical details to examine the effects of nucleosome breathing on the structure of small (<10 kb) chromatin systems. However, simulating chromatin arrays at high resolution (i.e., one bead per protein residue and DNA base pair) in physiological conditions is computationally challenging for various reasons. When represented at high resolution, chromatin is a high-dimensional system made up of a large number of oppositely charged particles (e.g., mainly lysine and DNA phosphate beads) that establish strong long-range interactions with each other. Such features give rise to a rugged energy landscape of chromatin that is populated by many competing low-lying minima separated by high energy barriers. A rugged energy landscape is difficult to sample with standard MD simulations, as transitions across the high energy barriers are rare within the accessible simulation timescales. Although Monte Carlo (MC) simulations are effective at overcoming high energy barriers, chromatin at high-resolution and under physiological conditions resembles a highly dense poly-branched polymer; consequently, most MC moves have a low acceptance probability due to steric clashes even after small displacements, just like in a dense liquid. To overcome these challenges and achieve sufficient sampling of chromatin at high resolution, we developed a Hamiltonian replica-exchange MD scheme that varies the Debye length across replicas. An advantage of this "Debye-length replica-exchange" approach, over standard temperature replica-exchange MD (T-REMD), is that it allows us to use a much smaller number of replicas (i.e., 16 instead of 80 for a 12-nucleosome system). This is achieved by implementing larger differences within the Hamiltonians of neighboring replicas but focused on selected degrees of freedom; such degrees of freedom are chosen to precisely modulate the electrostatic interactions, which contribute most strongly to the high energy barriers (Supplementary Methods). In addition, unlike the various temperature-replicas in T-REMD, each of the Debye-length-tuned replicas in our approach directly explores the behavior of chromatin at conditions easily accessible to experiments (e.g., each replica at a different salt concentration within the 0.01–0.15 mol/L NaCl range).

Using this Debye-length replica-exchange approach to sample our chemically-specific model, we can compare the behavior of chromatin with nucleosomes that exhibit spontaneous DNA unwrapping (i.e., nucleosome breathing) versus cases where the nucleosomes are constrained to remain permanently wrapped (i.e., no nucleosome breathing is allowed). As a benchmark for our model, we focus on 12-nucleosome chromatin arrays with a regular NRL of 165 bp (or a linker length of 18 bp); since in vitro sedimentation coefficients are available for validation[123], and because short regular NRLs favor folding into ideal zigzag structures well-characterized by near-atomic resolution in vitro experiments[13,14]. Furthermore, using a regular NRL across the array allows us to exclude the structural heterogeneity stemming from linker DNA variability[21], and to focus on the effects of nucleosome thermal breathing.

Our simulations reveal that constraining the nucleosomal DNA to remain fully and permanently attached to the histone core directs chromatin with short DNA linkers to fold into 30-nm rigid ladder-like zigzag fibers at 0.15 mol/L NaCl (see "Non-breathing" in Fig. 3a); i.e., where nucleosomes stack perfectly face-to-face with their second-nearest neighbors and rarely interact with other nucleosomes or in alternate orientations (top panel in Fig. 3b). The regular zigzag fiber structure we observe is analogous to that derived from cryo-EM experiments[14] for 12-nucleosome chromatin systems with short linker lengths, and the 167-bp tetranucleosome crystals[13].

More strikingly, our simulations show that the thermal breathing motion of nucleosomes destabilizes the formation of regular 30-nm zigzag fibers, and favors instead the organization of chromatin into a liquid-like ensemble ("Breathing" in Fig. 3a). This liquid-like ensemble encompasses a wide range of compact structures where nucleosomes interact with a multiplicity of neighbors in diverse orientations (i.e., face-to-face, side-to-side, and face-to-side) (bottom panel in Fig. 3b), as had been postulated by Maeshima and collaborators[18]. These ensembles are consistent with the disordered organization of nucleosomes observed with super-resolution nanoscopy[23] and chromEMT experiments of chromatin inside cells[16]. Furthermore, despite the high degree of variation in internucleosome interactions, the torsional and bending rigidity of the DNA intrinsically directs nucleosomes to engage in frequent, but orientationally diverse, interactions with their second-nearest neighbors. Therefore, chromatin forms a structure that lacks long-range order but that is underpinned by dominant interactions among $i$ and $i \pm 2$ nucleosomes, as observed by RICC-seq[27], Micro-C[25,26], and Hi-CO[28]. As a result of nucleosome disorder, liquid-like chromatin manifests a higher degree of compaction and flexibility than the 30-nm fiber. These features are evident from the wider distributions of the sedimentation coefficients (shifted towards the right) of chromatin with breathing nucleosomes, compared to those of chromatin with non-breathing nucleosomes (Fig. 3c). Importantly, the sedimentation coefficient predicted by our model for 165-bp chromatin at physiological salt (0.15 mol/L NaCl) is in quantitative agreement with the experimental value[123]. Our simulations also capture qualitatively the progressive decondensation of chromatin with decreasing monovalent salt observed experimentally[123].

To rationalize the impact of nucleosome breathing in chromatin self-assembly, we quantify the average number of DNA base pairs that unwrap from each nucleosome due to thermal fluctuations as a function of the NaCl concentration (Fig. 3d). In vitro, single-molecule (sm) fluorescence resonance energy transfer (FRET) experiments show that nucleosome breathing is hindered at 0.02 mol/L NaCl (i.e., ~10% probability; 1.4 ms unwrapped versus 14 ms wrapped), but is promoted as the salt concentration is increased to 0.1 mol/L NaCl (i.e., ~ 30% probability; 1.5 ms unwrapped versus 3–4 ms wrapped)[68]. Consistently, we observe that the average number of unwrapped DNA base pairs increases significantly with salt (i.e., from $7 \pm 2$ bp at 0.01 mol/L NaCl to $22 \pm 5$ bp at 0.15 mol/L NaCl). Such enhanced unwrapping at physiological salt concentrations originates from the weakening of the DNA–histone core attraction as the electrostatic screening increases. Notably, enhanced unwrapping of nucleosomes at physiological salt implies that the DNA linker regions continuously lengthen and shorten (i.e., the average length of the linker DNA doubles from 18 to ~38 bp). Beyond simply increasing the fluctuations in the internucleosome distances, dynamic variations in linker DNA length give rise to a very important concomitant effect: diversification of the rotational angles among immediately linked nucleosomes. In other words, because the DNA is a helix that twists 360° every ~10.5–10.7 bp[124], when the length of the DNA between two nucleosomes changes by a few bases, the nucleosomes are not only spaced out differently but also variably rotated with respect to one another. It is precisely the emergence of this marked

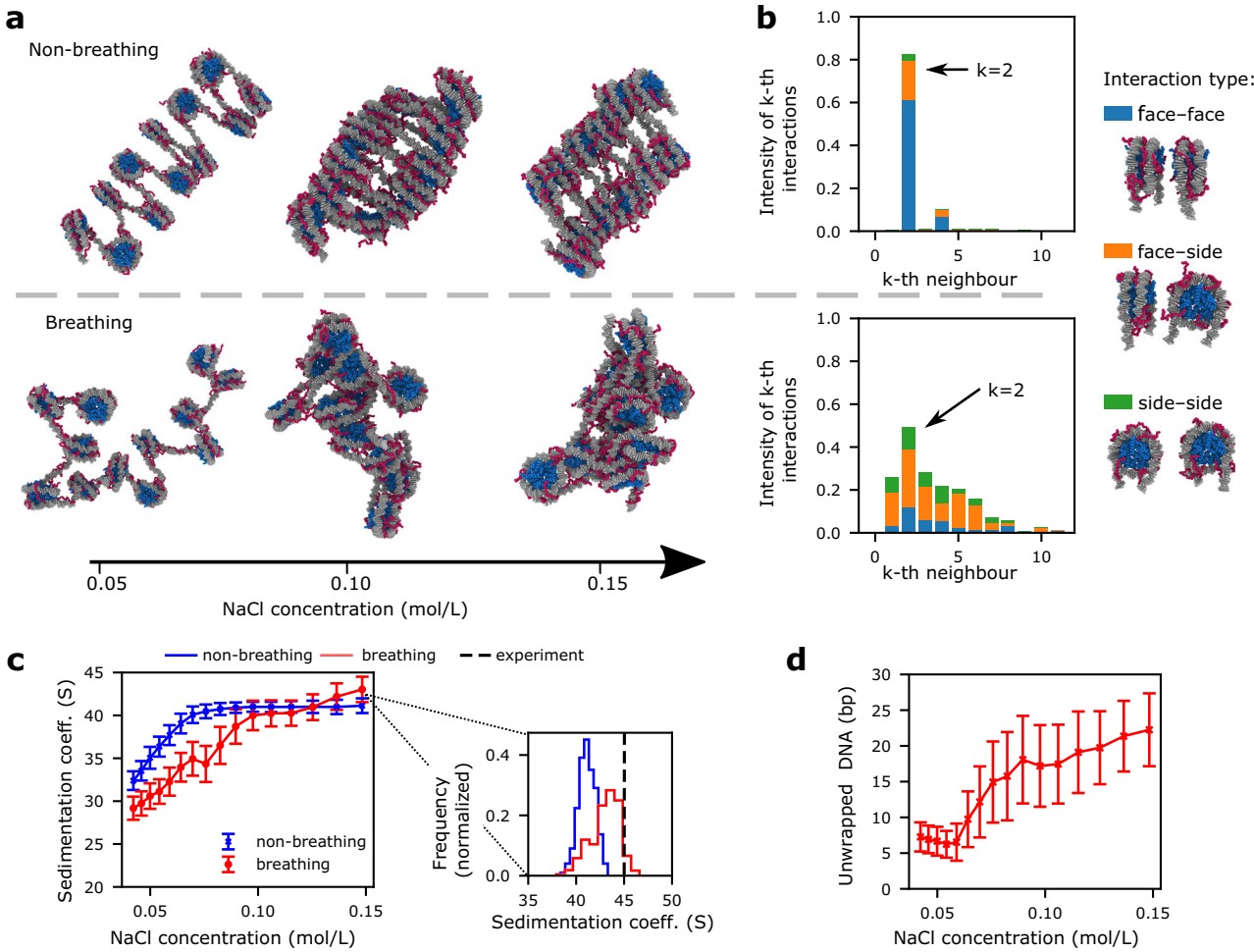

**Fig. 3 Structural differences in chromatin arrays with breathing versus non-breathing nucleosomes. a** Representative simulation snapshots of 165-bp 12-nucleosome chromatin with non-breathing (top) versus breathing (bottom) nucleosomes at three different salt concentrations: 0.05, 0.10, and 0.15 mol/L of NaCl (color-coded as in Fig. 1—level 2). **b** Bar plots depicting the frequency of interactions among $k$-th nearest nucleosomes neighbors for chromatin with non-breathing (top) versus breathing (bottom) nucleosomes. The bars are colored according to the percentage of the nucleosome pairs that engage in face-to-face (blue), face-to-side (orange), or side-to-side (green) interactions; these types of interactions are illustrated by the cartoons on the right. The definitions of the nucleosome axes used to determine if an interaction occurs face-to-face, face-to-side, or side-to-side are given in Supplementary Fig. 6. **c** Sedimentation coefficients versus NaCl concentration (left) for chromatin with non-breathing (blue) versus breathing (red) nucleosomes. The spread of the data around the mean values, mean ± s.d., are shown as bands; these were obtained by comparing $n = 500$ independent configurations. Histograms (right) comparing the distributions of sedimentation coefficient values for chromatin with non-breathing (blue solid) and breathing (red solid) at 0.15 mol/L in our simulations with the experimental value from reference[123] (black dashed). **d** Average number of DNA base pairs that unwrap per nucleosome in our simulations at a varying concentrations of NaCl. The spread of the data around the mean values, mean ± s.d., are shown as bands; these were obtained by comparing $n = 500$ independent configurations.

heterogeneity in internucleosome distances and rotational angles that explains, from a molecular point of view, the loss of long-range translational order in the organization of nucleosomes within liquid-like chromatin. Consistently, mesoscale simulations showed that large DNA linker variations increase the structural heterogeneity of chromatin[21], and experiments have shown that internucleosome rotational angle variability favors chromatin structural heterogeneity[31,123]. Moreover, mesoscale modeling revealed that binding of non-histone proteins, which can locally bend linker DNA, also destabilizes the regular 30-nm fiber folding[89]. The structural effects of nucleosome breathing that we observe are more pronounced in chromatin arrays with short NRLs, as the internucleosome orientations within such systems are otherwise highly restricted by their short linkers[123]. As the linker DNA lengthens, fluctuations become more energetically favorable and begin to intrinsically promote the heterogeneous nucleosome–nucleosome organization that sustains the liquid-like behavior of chromatin (Supplementary Fig. 11 and Supplementary Notes).

The modulation of nucleosome breathing with salt and its impact on chromatin structure may help explain why ordered and disordered structural chromatin models have been derived from in vitro and in vivo data, respectively. Such differences have already been attributed to the low salt concentrations used in many of the in vitro experiments[2,19], and to the regularity of reconstituted chromatin arrays—i.e., with strong nucleosome positioning sequences and homogeneous linker DNA sequences, uniform NRLs, homogeneous histone protein compositions, and a relatively small number of nucleosomes (~4–100)[21]. Our work further suggests that considering the disparities in the dynamic behavior of nucleosomes at physiological versus low salt is important in such debate.

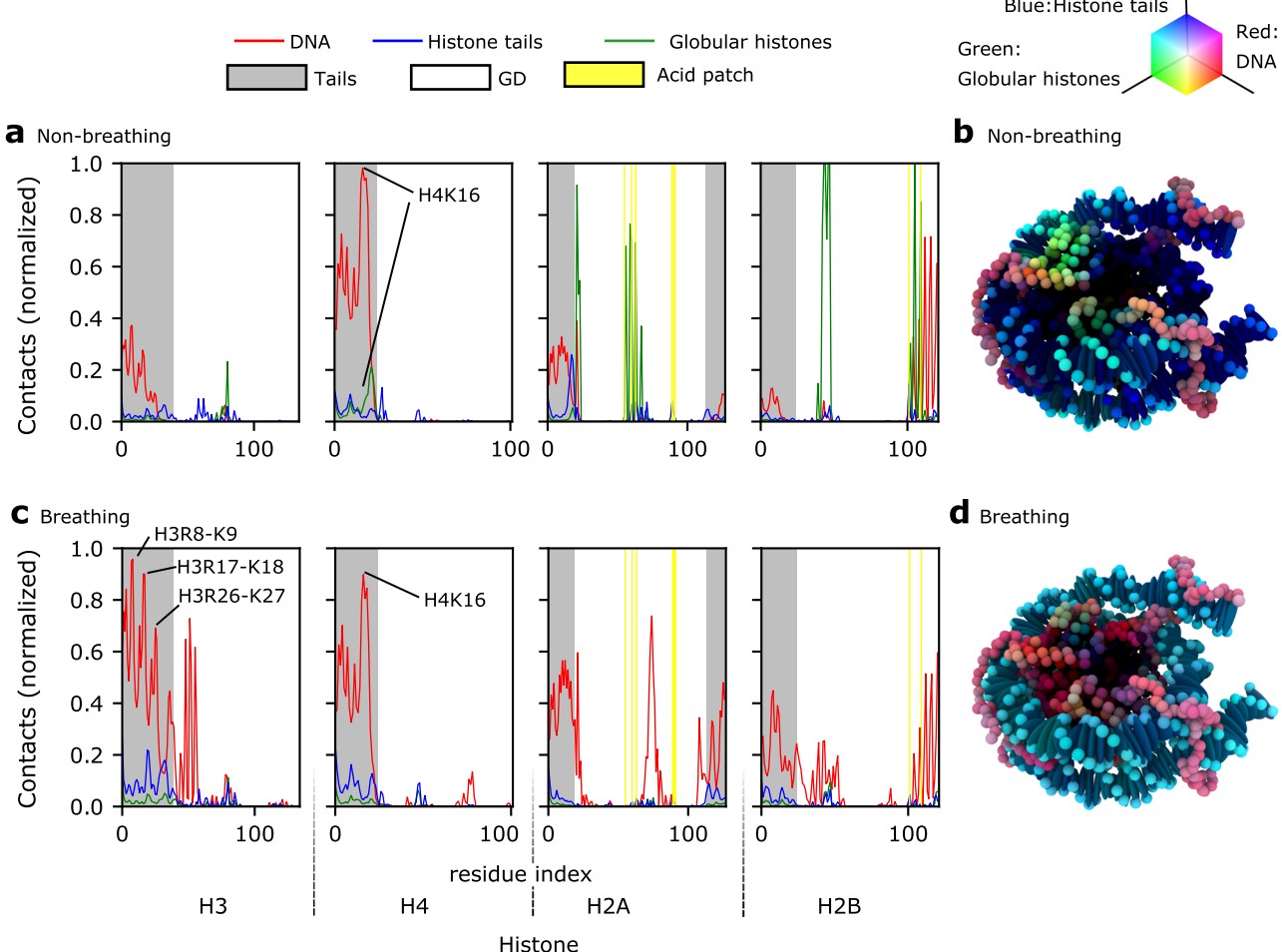

**Fig. 4 Molecular interactions that sustain chromatin compaction. a** Normalized fraction of nucleosome–nucleosome interactions (contacts) within compact chromatin for non-breathing nucleosomes that are mediated by histone–histone or DNA–histone interactions. The horizontal axes run across the histone protein residues within the core (i.e., H3, H4, H2A, and H2B, respectively) of a reference nucleosome, with the gray shaded areas highlighting histone tail residues, the white areas depicting globular domain residues, and the yellow vertical lines indicating the residues within the acidic patch for respective histones. The fraction of contacts is broken down by type: DNA–histone (red), histone tail–histone (blue), and globular domain–histone (green). **b** Visualization of the preferential types of interaction per residue or base pair in non-breathing nucleosomes. Each residue is represented by a sphere (centered on the $C_\alpha$) and each DNA base pair by two spheres (centered on the phosphates) and one ellipsoid. The particles are colored according to the RGB value that is obtained by combining the red, green, and blue values of lines in **a**. **c** Same as in **a** but for breathing nucleosomes. **d** Same as in **b** but for breathing nucleosomes.

## Liquid-like chromatin is stabilized by short-lived non-specific DNA–histone tail electrostatic interactions.

The ability of our chemically-specific model to resolve the motions of individual amino acids and DNA base pairs within compact chromatin enables us to examine the precise contributions of each of these species in directing chromatin organization. Specifically, we compute the fraction of time each amino acid or DNA base pair in a given nucleosome mediates internucleosome interactions. We categorize these interactions into three main groups: DNA-, globular histone-, and histone tail-mediated interactions. This analysis reveals that the molecular driving forces that stabilize the liquid-like organization of chromatin versus the regular 30-nm fiber folding are strikingly different (Fig. 4).

Electron microscopy and single-molecule force spectroscopy experiments on H4-tail cross-linked chromatin[125,126] show that folding of 30-nm zigzag fibers is driven by face-to-face interactions between the H4-tail of one nucleosome and the H2A histone on the surface of another. In agreement, we observe that 30-nm fibers are sustained by interactions between globular histones on the surfaces of the two stacked nucleosomes

(including those within the acidic patch), between the H4-tail and DNA, and more modestly between the H4-tail and the acidic patch (Fig. 4a, b). Besides the H4-tail—which has an ideal location on the nucleosome face—the other histone tails play a less prominent role in the folding of the 30-nm fiber. Our simulations further illuminate that the face-to-face stacking of nucleosomes with their second-nearest neighbors within zigzag fibers is long-lived (i.e., once stacked, nucleosomes rarely unstack; Fig. 5a).

A fascinating insight stemming from our simulations is that the molecular interactions sustaining the compact state of liquid-like chromatin are instead highly heterogeneous (Fig. 4c, d) and transient (i.e., nucleosomes bind and unbind dynamically; Fig. 5b); being most strongly contributed by non-specific electrostatic interactions between the various disordered histone tails (via their lysines and arginines) and the DNA (both nucleosomal and linker DNA). Hence, unlike in the 30-nm fiber, within liquid-like chromatin the important acid patch region is free to recruit a wide range of chromatin-binding factors present in cells[127]. Such diverse short-lived interactions are consistent with chromatin

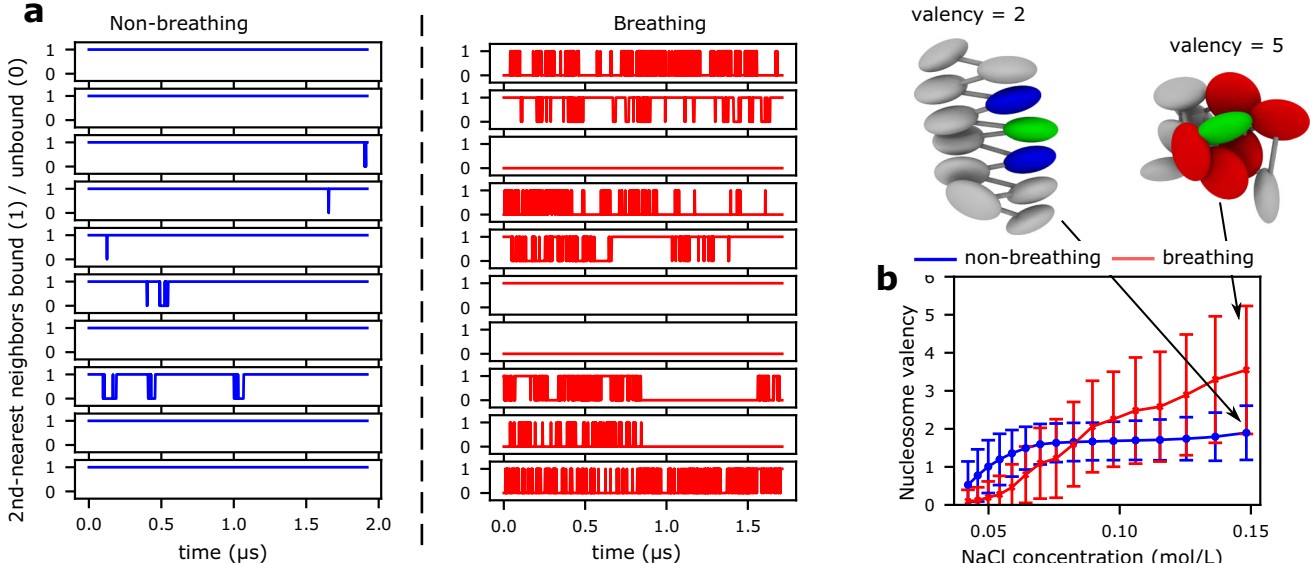

**Fig. 5 Dynamic nature of nucleosome–nucleosome interactions within compact chromatin. a** Time series of the binding (1) and unbinding (0) of second-nearest nucleosome neighbor pairs computed from ten independent unbiased MD trajectories at 0.10 mol/L NaCl. The binding/unbinding behavior of non-breathing (left; blue) versus breathing (right; red) nucleosomes is summarized. **b** Average valency of a non-breathing (blue) versus a breathing (red) nucleosome—defined as the average number of nucleosome neighbors that each nucleosome "contacts" simultaneously (i.e., within a center-to-center distance of 110 Å; see Supplementary Methods)—within compact chromatin versus NaCl concentration. The spread of the data around the mean values, mean ± s.d., are shown as bands; these were obtained by comparing $n = 500$ independent configurations. The cartoon at the top-left highlights a non-breathing reference nucleosome (green) with a valency of two (i.e., in simultaneous contact with the two blue nucleosomes). The cartoon at the top-right illustrates a breathing nucleosome (green) that is in simultaneous contact with five other nucleosome neighbors (red).

dynamically transitioning between the heterogeneous compact structures that make up the liquid-like ensemble. The key importance of histone tail–DNA electrostatic interactions is supported by experiments demonstrating that chromatin with tail-less nucleosomes fails to condense[128]. In addition, mutation of all H4 arginine and lysine residues to alanine, or histone tail acetylation inhibits nucleosome–nucleosome interactions and the intrinsic LLPS of chromatin[43].

Interestingly, despite the huge differences in the molecular interactions that stabilize the liquid-like and the fiber-like structures of chromatin, the dominant role of H4K16 is uncontested in both cases (Fig. 4). Among the contacts established by the H4-tail, the electrostatic interactions mediated by K16 are the strongest (most significantly with DNA in both types of chromatin structures, and more modestly with the acidic patch in the zigzag fibers). This dominance of the H4K16 residue is in agreement with the well-known decompaction triggered by H4K16 acetylation[128,129], and the observation that reversible acetylation of H4K16—one of the most frequent post-translational modifications across organisms—has diverse functional implications[130].

Besides characterizing the molecular features of liquid-like chromatin, we were interested in understanding the underlying physical principles that drive chromatin to adopt such a disordered organization, in terms of its thermodynamics. The structural heterogeneity of the liquid-like ensemble (Fig. 3c) indicates that such organization decreases the free energy of chromatin by expanding the number of accessible microstates (entropy gain), compared to those available in a regular 30-nm fiber. Next, to assess the variation in enthalpy, we define the "valency" of a nucleosome as the average number of nucleosome–nucleosome contacts it establishes (Supplementary Methods). We observe that nucleosomes within liquid-like chromatin have a significantly higher valency at physiological conditions than nucleosomes within 30-nm fibers (Fig. 5b).

Hence, beyond the entropic driving force, nucleosomes within a liquid-like ensemble decrease their free energy by establishing numerous, but still transient, attractive interactions that maximize the enthalpy gain upon chromatin compaction. Numerous weak and transient internucleosome contacts are preferred over fewer strong longer-lived face-to-face interactions, as the latter represents a greater entropic cost.

**Physical and molecular determinants of intrinsic liquid–liquid phase separation of chromatin.** Recent groundbreaking experiments discovered that 12-nucleosome reconstituted chromatin undergoes intrinsic LLPS—i.e., without the aid of additional proteins—under physiological salt concentrations in vitro and when micro-injected into cells[43]. Extensive studies characterizing the LLPS of proteins and nucleic acids have demonstrated that multivalency is the dominant driving force for their LLPS[58–60,131–133]. That is, proteins with high valencies can stabilize LLPS by forming numerous weak attractive protein–protein[58,59,131], protein–RNA[134], and/or protein–DNA[40,41] interactions that compensate for the entropy loss upon demixing[60]. Furthermore, binding of multiple Swi6 (the Schizosaccharomyces pombe HP1 protein) molecules to nucleosomes was recently shown to reshape the nucleosome in a manner consistent with nucleosome breathing—i.e., increasing the solvent exposure of core histones—and subsequently promote HP1-chromatin LLPS[52]. These ideas, together with our observations of nucleosome valency enhancement from spontaneous breathing, led us to hypothesize that nucleosome plasticity could be crucial in facilitating the intrinsic LLPS of chromatin.

To investigate this phenomenon and gain molecular and thermodynamic insight, we use our minimal coarse-grained chromatin model, as it simultaneously predicts chromatin ensembles in quantitative agreement with those of our chemically-specific model (Supplementary Fig. 4) and can simulate a solution of hundreds of interacting chromatin arrays. Specifically, we perform direct coexistence simulations of systems containing 125

independent 12-nucleosome chromatin arrays with a uniform NRL of 165 bp at different conditions. The direct coexistence method involves simulating two different phases—the condensed (chromatin-enriched) liquid in contact with the diluted (chromatin-depleted) liquid—in the same simulation box separated by an interface[135–137] (Supplementary Methods). From these simulations, we compute full liquid–liquid phase diagrams of chromatin at constant room temperature in the "NaCl concentration" versus "chromatin density" space (Supplementary Methods and Supplementary Fig. 5), and compare the results for chromatin with breathing nucleosomes and, as a control, for nucleosomes that are artificially constrained to remain fully wrapped (i.e., non-breathing).

Our phase diagrams allow us to compare the conditions under which chromatin LLPS takes place spontaneously for the two representations. This analysis reveals that, besides promoting the liquid-like behavior of individual chromatin arrays, nucleosome breathing increases the range of stability of chromatin LLPS by enhancing the valency of nucleosomes (Fig. 6a). That is, chromatin spontaneously forms condensates above a critical monovalent salt concentration; i.e., where screening by counterions is sufficiently strong to eliminate the DNA–DNA repulsion and encourage the formation of numerous weak and transient attractive internucleosome interactions. Compared with breathing nucleosomes, when nucleosomes are constrained to remain fully wrapped, chromatin phase separation requires higher NaCl concentrations to become thermodynamically stable (Fig. 6a); the limited valency of non-breathing nucleosomes necessitates the formation of stronger nucleosome–nucleosome interactions to obtain sufficient enthalpic gain for LLPS. Beyond the critical solution salt concentration, the size of the liquid–liquid coexistence region is significantly larger for chromatin with nucleosomes that breathe spontaneously (Fig. 6a); i.e., under the same solution conditions, breathing nucleosomes yield a more dense and, hence, more stable condensed liquid.

The physical forces governing LLPS of chromatin can be analyzed by computing the average density of molecular connections (i.e., bonds) that nucleosomes form per unit of volume within the condensed phase as a function of the salt concentration. Similar to its effect in enhancing the valency of nucleosomes within single chromatin arrays, we observe that nucleosome breathing promotes LLPS because it fosters a much higher nucleosome–nucleosome connectivity within the condensed liquid (Fig. 6b). A densely connected chromatin liquid phase fulfills two crucial requirements: making LLPS thermodynamically favorable and ensuring that the condensate mantains liquid-like properties. For instance, a condensed chromatin liquid and a gel would exhibit a similar percolating network structure (i.e., where each chromatin array is bound to at least one other array). However, while chromatin arrays diffuse freely within liquids, they are dynamically arrested within gels, as a result of increasingly strong and long-lived nucleosome–nucleosome interactions. Therefore, to preserve the liquid-like properties of the chromatin condensate, nucleosome–nucleosome interactions must remain sufficiently weak, and thus short-lived. Importantly, these weak nucleosome-nucleosome interactions must establish a densely connected liquid network to compensate for the entropy loss upon demixing, and make LLPS thermodynamically stable. Hence, liquid chromatin condensates are favored by the dynamic formation and rupture of a large number of weak attractive nucleosome–nucleosome interactions, which in turn are facilitated by the dynamic breathing of nucleosomes.

## Discussion

We introduce a mechanistic multiscale model of chromatin designed to investigate the connection between the fine-atomistic details of nucleosomes and the emergence of chromatin self-organization and LLPS in systems with over a thousand nucleosomes. By combining atomistic simulations, a residue/base-pair resolution model, and a minimal representation of chromatin, our multiscale approach enables the study of collective effects of amino acid sequence and mutations, post-translational modifications, histone secondary structural changes, DNA sequence, and nucleosome dynamics in the modulation of the mesoscale structural properties of chromatin.

Our simulations put forward nucleosome plasticity as a key driving force of the intrinsic liquid-like behavior of chromatin at physiological salt concentrations. In vivo, nucleosome plasticity can originate not only from nucleosome breathing, but also from nucleosome sliding, binding of proteins like HP1[52], nucleosome remodeling, post-translational histone modifications, histone replacement, and other mechanisms. We observe that nucleosome plasticity transforms nucleosomes from the uniform and static disc-like repeating units needed to sustain rigid zigzag fibers, to highly heterogeneous and dynamical particles that engage in promiscuous nucleosome–nucleosome interactions, sample a wide range of internucleosome rotational angles, and spontaneously self-assemble into disordered structures. Such a disordered organization of nucleosomes is in agreement with the liquid-like model of Maeshima and colleagues[2,18,19], and the ultrastructure of in vivo chromatin visualized by chromEMT[16] and super-resolution nanoscopy[23].

Regardless of the marked heterogeneity and promiscuity of internucleosome interactions, we find that nucleosomes within liquid-like chromatin engage in frequent $i$ and $i \pm 2$ contacts, as observed in sequencing-based experiments of chromatin in situ[25–28]. However, unlike those sustaining zigzag fibers, the dominant second-nearest neighbor interactions found within liquid-like chromatin are highly orientationally diverse and transient, and do not resemble a stack of tetranucleosomes. Furthermore, in contrast to the 30-nm zigzag fibers, which are mostly assembled via long-lived face-to-face H4-tail to acidic patch interactions, liquid-like chromatin is instead stabilized by a diverse range of short-lived non-specific DNA–histone tail electrostatic interactions. Therefore, within liquid-like chromatin, the nucleosome acidic patch region can be more easily accessed by the myriad of chromatin-binding factors that have been proposed to target it in vivo[127,138], which may include HP1[139]. Furthermore, controlled access to the acidic patch region has been hypothesized to play a crucial role in the modulation of chromatin remodeling motors that regulate nucleosome sliding[140].

We also find that the attractive interactions that maintain the DNA wrapped around the histone core are predominantly electrostatic in nature. Hence, nucleosome breathing is boosted by electrostatic screening at physiological salt concentrations and hindered at lower salt concentrations, as also shown experimentally[68]. Our work further reveals that significant nucleosome breathing at physiological salt favors the liquid-like behavior of chromatin even in artificially homogeneous oligonucleosomes (e.g., with uniform DNA linker lengths and DNA sequences and regular histone compositions). In contrast, lowering the salt concentration, as done in some in vitro experiments, inhibits nucleosome fluctuations and drives chromatin to form ordered 30-nm zigzag fibers. Accordingly, the modulation of nucleosome breathing with salt may aid in reconciling discrepancies between the ordered and disordered chromatin structural models derived from in vitro and in vivo experiments, respectively.

Importantly, our work demonstrates that liquid-like chromatin is simultaneously stochastically organized and tightly packed. This suggests that chromatin compaction does not immediately imply steric barring of enzymes and that indeed chromatin as a compact liquid-like system is optimum for DNA searchability;

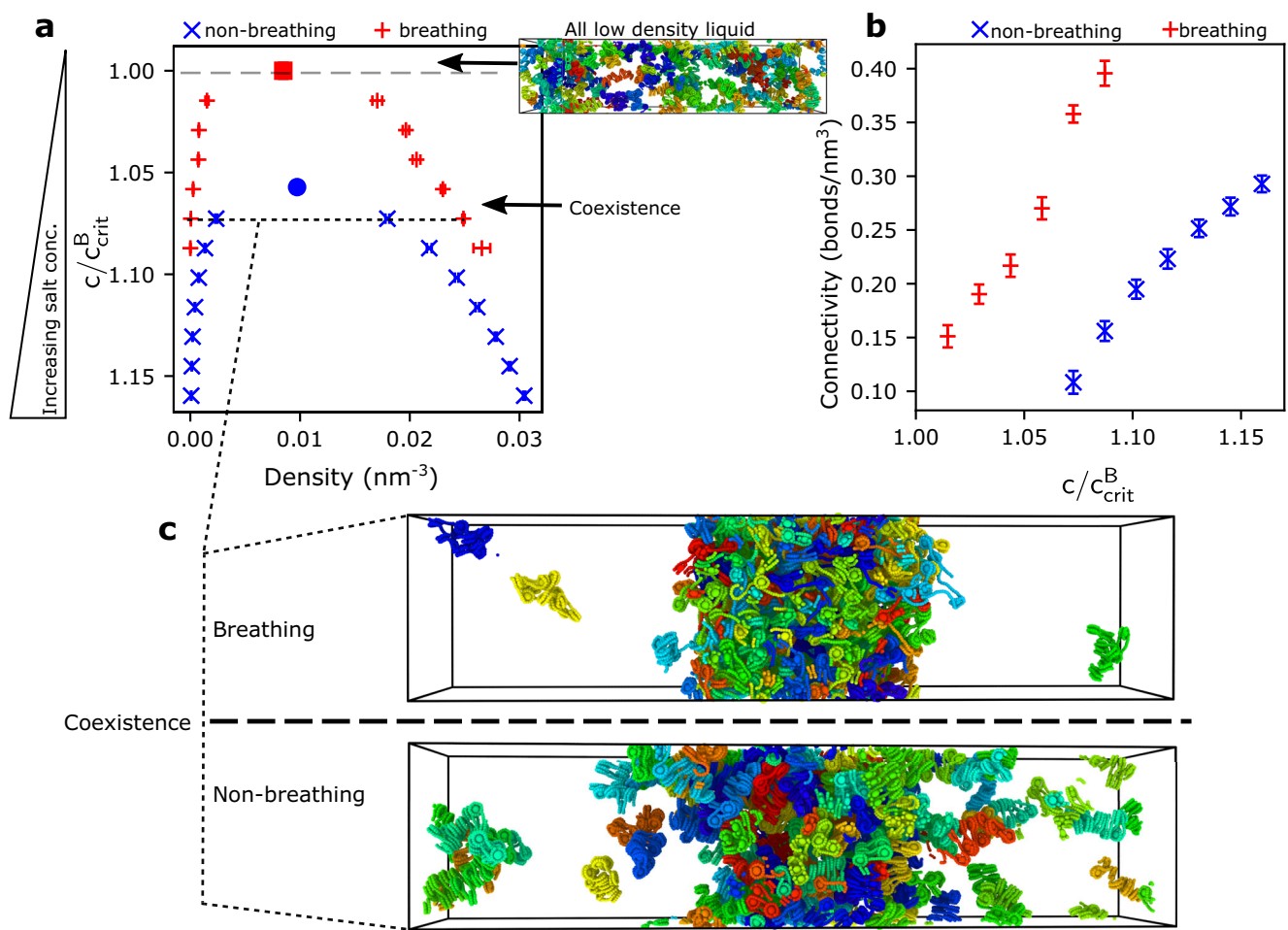

**Fig. 6 Impact of nucleosome breathing on the phase behavior of chromatin. a** Phase diagram of a solution of 12-nucleosome chromatin arrays exploring the space of decreasing concentration of NaCl (vertical axis, $C$) versus density (horizontal axis). The data points (blue: non-breathing, red: breathing) represent coexistence points, i.e., the densities of the chromatin-diluted (left branch) and chromatin-enriched (right branch) coexisting liquid phases at a given monovalent salt concentration. The density is defined as the number of chromatin molecules per unit of volume in $nm^{-3}$. Statistical errors, mean ± s.d., are shown as bands; these were obtained from the density profile curves (Supplementary Fig. 5) by comparing $n = 100$ density values within the plateau characterizing each phase. The region above the coexistence curve is the "one-phase region", where the NaCl concentration is too low to screen DNA–DNA electrostatic repulsion and enable the attractive nucleosome–nucleosome interactions, needed for LLPS (see top snapshot of a well-mixed low-density liquid). A representative simulation snapshot of the well-mixed low-density liquid for chromatin with non-breathing nucleosomes is shown. Each independent 12-nucleosome array within the solution is colored differently to aid visualization; the DNA beads are shown as small spheres and each nucleosome histone core with one larger sphere (as described in Fig. 1—level 3). The region below the coexistence curve represents the "two-phase region" where chromatin demixes into a condensed (chromatin-enriched) and a diluted (chromatin-depleted) liquid phase. The maximum in the coexistence curve represents the critical point: if the NaCl concentrations exceed the critical value, LLPS occurs spontaneously. The vertical axis has been normalized by the critical salt concentration for chromatin with breathing nucleosomes ($C/C_{crit}^B$), where $C_{crit}^B = 0.069$ mol/L of NaCl. The critical salt concentration for chromatin with non-breathing nucleosomes is $C_{crit}^{NB} = 0.073$ mol/L. **b** Connectivity of the condensed liquid formed by chromatin with non-breathing (blue) versus breathing (red) nucleosomes. The connectivity is defined as the mean number of connections that chromatin arrays within the condensed liquid form (i.e., the number of distinct chromatin arrays that a reference array is in contact with, considering any two nucleosomes within a center-to-center distance shorter than 110 Å) multiplied by the density of the condensed phase. Statistical errors, mean ± s.d., are shown as bands; these were obtained by comparing the values among from $n = 300$ independent configurations. **c** Representative simulation snapshots of the phase-separated liquids formed by chromatin with breathing (top) versus non-breathing (bottom) nucleosomes at a value of $C/C_{crit}^B = 1.073$ (i.e., $C = 0.074$ mol/L of NaCl). The 12-nucleosome arrays in the snapshot are depicted as described in **a**.

i.e., liquid-like chromatin may provide easier and more homogeneous DNA access for processes like "scanning and targeting genomic DNA" without the need for chromatin to undergo decompaction, as postulated by Maeshima and collaborators[19].

The strong link between the stochastic organization of chromatin—which we show might facilitate access to nucleosomal DNA—and nucleosome plasticity might have important functional implications. Nucleosomes represent a fluctuating barrier for the binding of transcription factors to DNA, and hence, for

transcription[75,76]; the maximally repressive state is that of the fully wrapped nucleosome, while the maximally non-repressive state is that of a nucleosome-free region[141]. However, although the majority of transcription factors seem to bind to nucleosome-free DNA regions[142,143], spontaneous nucleosome breathing may provide transient access to some of these proteins to their nucleosome binding sites and, hence, facilitate transcription initiation[144–146], especially at physiological ionic strengths[69]. The increased access of chromatin due to nucleosome breathing might

be most relevant for rationalizing the mechanisms of binding of pioneering factors[147], which are a special class of proteins that target nucleosomal DNA and in compact chromatin[148,149]. Additionally, in vitro, nucleosome breathing has been suggested to play a role in transcription elongation, by facilitating the movement of RNA polymerase II ternary elongation complex across the nucleosomal DNA[75,150]. In vivo, spontaneous nucleosome breathing, combined with the action of chromatin remodellers, allows the Clustered Regularly Interspaced Short Palindromic Repeats (CRISPR)-associated protein 9 (Cas9) to access the nucleosomal DNA[151]. Our observations also support one potential mechanism influencing the control of gene transcription kinetics: spontaneous thermal breathing. This breathing behavior has been proposed to allow nucleosomes to adopt a wide range of promoter configurations, some of which transiently facilitate transcription and others that momentarily inhibit it; hence, modulating the rate of discontinuous transcription of genes that includes bursts of activity[141]. Our work strongly suggests that enhancement of nucleosome breathing at promoters would indeed foster their structural heterogeneity and favour LLPS.

Significantly, we demonstrate that the same molecular and biophysical driving forces that sustain the liquid-like behavior of nucleosomes within single compact chromatin arrays also promote the intrinsic phase separation of a solution of chromatin arrays. Specifically, nucleosome breathing fosters not only the formation of disordered and flexible chromatin ensembles but also LLPS and the emergence of phase-separated chromatin compartments. Interestingly, when nucleosome breathing is artificially suppressed (analogous to the behavior of strongly positioning artificial sequences), chromatin shows a reduced propensity for LLPS. Thus, intranuclear conditions that can spontaneously tune intrinsic nucleosome breathing via modulation of electrostatic interactions (e.g., changes in ionic salt concentration, pH, DNA/histone mutations) are expected to alter the spatial organization, degree of compaction, and compartmentalization of chromatin. Modulation of nucleosome breathing may, therefore, represent a key mechanism used by cells, not only to organize chromatin but also, to modulate its function.

Our work also reveals that nucleosome breathing and multivalency are intricately linked and are positively correlated. Multivalency has been previously identified as an important property governing protein LLPS, both in the cytoplasm and nucleoplasm. Furthermore, protein multivalency and, therefore, LLPS can be modulated by several external (e.g., salt concentration, temperature, pH, multi-component composition) and intrinsic factors (e.g., protein mutations, post-translational modifications, and disorder-to-order transitions). For chromatin, we show that an increase in the monovalent salt concentration facilitates nucleosome breathing, and subsequently, enhances nucleosome multivalency. Moreover, our work strongly suggests that, in the absence of significant environmental changes, within a fiber-like chromatin model, the valency of nucleosomes can be considered as roughly static; i.e., for short linker DNAs, the torsional rigidity of the DNA locks second-nearest nucleosome neighbors in an arrangement where they bind exclusively to one another and rarely to other nucleosomes. In contrast, within a plastic nucleosomal framework, nucleosomes, like proteins, possess an inherent capacity (i.e., nucleosomal breathing) to dynamically modify their valency, and ultimately regulate their functionality.

Recent landmark experiments by Sanulli et al.[52] demonstrated that the binding of many HP1 molecules unexpectedly reshapes nucleosomes and makes the histone core more accessible to the solvent; consistent with nucleosomal DNA unwrapping, which in turn can increase the availability of the core for interactions that facilitate LLPS. Our work further suggests that the increase in the plasticity of nucleosomes induced by HP1 is consistent with the amplification of: (a) the local flexibility of chromatin, (b) the range of accessible nucleosome–nucleosome pair orientations, (c) the effective valency of nucleosomes, and (d) the transient nature of the inter-nucleosomal attractive interactions that lead to chromatin LLPS.

Together, our work postulates that nucleosome plasticity is an intrinsic property of nucleosomes that facilitates chromatin stochastic self-assembly and LLPS, and hence, contributes to regulating the organization and membrane-less compartmentalization of the genome. These findings advance the molecular mechanisms and biophysical understanding of how the liquid-like organization of the genome is formed and sustained, and how it can be regulated. Modulation of nucleosome plasticity might have important implications in the functional organization of the genome and in the control of gene transcription parameters.

## Methods

**Multiscale approach.** In this work, we develop a bottom-up multiscale modeling technique—combining atomistic representations (Level 1) with two levels of coarse-graining (Levels 2 and 3)—to link the chemical heterogeneity of chromatin and the spontaneous breathing motions of nucleosomes to chromatin self-assembly and its LLPS. We have chosen a multiscale strategy because it allows us to take advantage of (1) the ability of atomistic models to elucidate how chemical changes transform the local behavior of proteins and DNA, describe the binding of proteins to chromatin, and establish how DNA sequence transforms its mechanical properties, and (2) the capacity of coarse-grained descriptions to reduce the system dimensionality markedly (e.g., represent a 300 kb chromatin region, or about 30 million atoms plus solvent, with as little as ~50,000 beads). The need for combining two levels of coarse-graining, instead of just one, stems from our interest in describing systems with thousands of nucleosomes (Level 3) while retaining essential physicochemical information mapped from the all-atom level (our intermediate resolution, or level 2). The two interconnected levels of coarse-grained models we have developed are described below along with the rationale behind their designs. An animated illustration of our multiscale strategy can be found in https://sef43.gitlab.io/.

**Chemically specific chromatin coarse-grained model.** When designing our chemically-specific coarse-grained model (Level 2), our goal was to fulfill two opposing requirements. The first requirement was to describe proteins and DNA at high-enough resolution to capture the effects of DNA and amino-acid sequence variations in nucleosome–nucleosome interactions, and also the intrinsic nucleosome breathing motions that emerge naturally from the histone–DNA interactions and the mechanical properties of the DNA. The second requirement was to reduce the number of degrees of freedom within chromatin as much as possible to efficiently simulate oligonucleosome systems. Describing proteins at a resolution of one bead per amino acid, and the DNA at a resolution of one bead per base pair, although expensive computationally, is the ideal choice to realize these two goals; i.e., such a resolution is needed to retain the full chemical composition of a heterogeneous chromatin array and map, from the bottom up, the diverse physicochemical properties of the distinct amino acid and nucleobases that make up each of the different nucleosomes within chromatin. This resolution is also needed to consider amino acid point mutations, if desired, interrogate the role of specific histone residues in mediating chromatin organization, probe binding of additional proteins to specific chromatin residues, and describe the contributions of distinct amino acids to nucleosome unwrapping. Furthermore, while histone core proteins are largely α-helical and exhibit relatively small structural fluctuations in crystallographic studies[3,7] and in atomistic MD simulations[38], histone tails are largely disordered and highly flexible. A resolution of one bead per residue is the coarsest resolution that can describe the detailed topology of the histone globular domains and the flexibility of the histone tails, and consider disorder-to-order transitions that might be triggered by post-translational modifications[129]. In terms of DNA, a resolution of one bead per base pair is the coarsest resolution that can capture in full the influence of DNA sequence on the mechanical properties of DNA (i.e., twist, roll, and twist). Collectively, our chemically-specific model reduces the number of particles in a 5 kb chromatin region, or in a ~25 nucleosome system, from ~0.5 million atoms (plus solvent) to only ~25,000 beads. Below, we provide more details of this model.

Following the work of Dignon and colleagues[105], we represent each amino acid explicitly—both within the histone globular regions and histone tail regions— by using a single bead that carries the charge, hydrophobicity, and size of its atomistic counterpart. Each bead is defined as a point particle with an excluded volume centered on the $C_\alpha$ atom of the amino acid it represents. For each amino acid, the bead diameter is calculated from the experimentally measured van der Waals volumes and assuming that amino acids have a spherical shape, as done previously[103]. We also assign a charge to each bead corresponding to the total

charge of the related amino acid. The sequence-dependent hydrophobic attraction between specific amino acid pairs is accounted for by the Kim–Hummer model[103,105], which consists of a shifted and truncated Lennard–Jones potential with parameters derived from experimental amino-acid pairwise contact propensities[104]. Using our previous microsecond-long Bias-exchange metadynamics molecular dynamics simulations of 211-bp nucleosomes[38], we differentiate between the globular histone core, residues that retain their secondary structure throughout the simulation, and the disordered histone tails. We then treat amino acids within intrinsically disordered regions as fully flexible polymers (i.e., with no energetic penalty for bending) using a harmonic potential with a stiffness bond constant $k_b$ of 20 kcal/mol/Å$^2$ and a resting length $r_0$ of 3.5 Å (Supplementary Methods), as proposed by Dignon and colleagues[105]. We describe the relatively small structural fluctuations within the histone globular domains by building an elastic network model[152], which avoids the need of including internal non-bonded terms in these regions. In practice, we take a representative structure from the highest populated cluster in our atomistic simulations[38] as the reference structure and connect all the globular histone core beads that are within 7.5 Å of each other with harmonic springs and form a Gaussian elastic network model (GNM)[152]. For the harmonic bond interaction of the GNM, we use a spring constant $k_{GNM}$ of 20 kcal/mol/Å$^2$, and the equilibrium distance among amino-acid pairs is set equal to its value in the reference atomistic structure. Linker histones, and other chromatin-binding proteins of interest, can be described with our model in the exact same way as histones. That is, first, an initial coarse-grained model is built starting from a high-resolution structure of the protein (e.g., from ref. [153] and adding the intrinsically disordered regions as described in ref. [38] for H1). Next, using either experimental structural data or atomistic simulations, the degree of order/disorder in the protein regions is defined. Finally, each amino acid is represented with one bead, and residues within globular domains are connected by an elastic network model, while those within disordered regions are described as fully flexible chains.

The second crucial feature of our model is that it considers sequence-dependent DNA mechanical properties by using the rigid base pair (RBP)[98–102] model with added phosphate charges. The RBP model represents one DNA base-pair step with one single ellipsoid and depicts the DNA conformational changes in terms of harmonic deformations of six helical parameters (three angles: twist, roll, and tilt, and three distances: slide, shift, and rise) that account for the relative orientations and positions of neighboring base-pair planes. The DNA mechanical potential energy is computed from the sum of harmonic distortions of equilibrium base-pair step geometries. We used the Orozco group parameters[101,102] computed from MD atomistic simulations—i.e., the equilibrium values by fitting Gaussian functions to the distributions of helical parameters, and the elastic force constants by inversion of the covariance matrix in helical space. In practice, our model represents single base pairs, both within the nucleosomal and linker DNA, by one coarse-grained bead defined by a position vector, $\mathbf{r}$, and an orientation quaternion, $q$. We add two virtual charge sites to each DNA ellipsoid (i.e., one per phosphate approximating the shape of the DNA phosphate backbone) to consider the crucial electrostatic interactions that drive chromatin self-organization. We implement this RBP plus charged virtual sites in LAMMPS (http://lammps.sandia.gov)[154] with an ellipsoid defined by two-point particles with a negligible but non zero mass. While the combined three-particle base-pair bead is treated as a single rigid body for the dynamics, the individual components each contribute to the calculation of the potential and forces.

Besides the excluded volume and hydrophobic non-bonded interactions, we consider electrostatic interactions between the charged beads by means of a Debye–Hückel potential, but omit all non-bonded interactions between directly bonded beads. The binding of the nucleosomal DNA to the histone protein core is achieved through these protein–DNA electrostatic and hydrophobic interactions, resulting in a nucleosomal DNA that wraps ~1.7 times around the histone core and exhibits spontaneous unwrapping/re-wrapping. The force-induced unwrapping behavior of these nucleosomes is in quantitative agreement with experiments at single base-pair resolution (see "Model validation" section and the Supplementary Notes). To model nucleosomes that are artificially constrained to be "non-breathing", we describe the histone core together with the bound nucleosomal DNA as a single GNM, using the same 7.5 Å threshold and bond parameters. This results in the nucleosomal DNA being constrained to remain permanently bound to the histone core, and inhibits both nucleosome breathing and sliding.

The Debye–Hückel approximation is a mean-field theory where the ion density is given by the linearized Boltzmann distribution and effects like ion condensation, ion correlations, ion heterogeneity, and specific ion binding are ignored. By invoking such approximation, we assume that the effects of monovalent counterions in the solution can be reduced to simply screening the mean electrostatic potential from the chromatin charges, and as such, we describe the decay of charge–charge interactions with distance by a Yukawa function. Such an approximation is crucial to reduce the high dimensionality of the chromatin phase space and to enable us to sample it efficiently. Importantly, this approximation is exact in the low salt limit, and previous chromatin coarse-grained models have shown that it captures well the salt-dependent compaction of chromatin at the low to moderate monovalent salt concentrations present inside cells (≤0.15 mol/L of NaCl)[21,81]. However, its numerical implementation becomes progressively challenging as the salt concentration reaches very low values because estimating the interactions requires larger and larger Debye lengths and, hence, larger and larger cutoffs (Supplementary Methods).

An additional limitation of invoking the Debye–Hückel approximation is that it neglects the effects of $Mg^{2+}$ ions in solution, which are highly abundant inside cells (i.e., ~10–20 mmol/L but with the fraction of free $Mg^{2+}$ ions estimated at <5% of this[155]). $Mg^{2+}$ ions are thought to outcompete monovalent ions and interact preferentially with the multiple closely-spaced negative charges on the nucleosome surface and on the DNA[156]. Reassuringly, at such low concentrations, $Mg^{2+}$ ions have been shown to only modestly enhance chromatin compaction in vitro, e.g., by ~2% and ~13% when comparing the sedimentation coefficients of a 165-bp 12-nucleosome array in 0.15 mol/L NaCl versus in 1 and 2 mmol/L MgCl$_2$, respectively[123]. Importantly, landmark work of the late Jonathan Widom showed that in the presence of 0.1 mol/L Na$^+$, the addition of 0–10 mmol/L Mg$^{2+}$ did not substantially change the equilibrium constant for nucleosome unwrapping[145]. More recent sm-FRET studies observed that increasing the ionic strength of the buffer, by raising instead the NaCl concentration to up to 0.4 mol/L, had a minimal impact on nucleosome unwrapping[157]. Hence, the effects that increased ion screening by Mg$^{2+}$ ions at physiological concentrations are likely subtle and expected to take place at much finer scales than what we set to capture with our chemically-specific model. Furthermore, we note that the experimental data that we have used to quantitatively validate the force-induced unwrapping response of our model (i.e., mechanical unzipping experiments of Wang and colleagues[120]) is for nucleosomes already immersed in a buffer containing both 0.1 mol/L NaCl and 0.5 mmol/L MgCl$_2$, suggesting that the balance of parameters in our model captures well this regime.

In summary, our chemically-specific coarse-grained model of chromatin preserves the atomistic shape and size of the nucleosome core, the length and flexibility of the histone tails, the explicit charges and hydrophobic nature of all the different amino acids within the histone protein (including those in the important acid patch region[3]), the sequence-dependent mechanical properties of the DNA, and the thermal breathing motions of nucleosomes. A list of all the parameters we use in our chemically-specific model, along with the choice of values and justification is given in Supplementary Table 9.

**Minimal chromatin coarse-grained model**. In a similar fashion to the design of our chemically-specific model, our goal when developing the minimal model (Level 3) was to find the coarsest possible representation of nucleosomes that simultaneously allows us to (1) account for the breathing motions of nucleosomes and quantitatively reproduce the chromatin structural ensembles that we observe with our chemically-specific model, and (2) model chromatin systems with thousands of nucleosomes to reach functionally relevant scales or phase-separating systems. To map the breathing motions of nucleosomes from our chemically-specific simulations, and also the torsional and bending properties of the linker DNA (which we find crucially determine the configurational ensembles of chromatin), we require an explicit representation of the DNA—this is indeed one of the limiting factors defining the resolution of our minimal model. The exact resolution of our minimal DNA model is dictated by the physical diameter of a canonical B-DNA strand and the type of potentials we use to describe interactions among minimal DNA beads. Specifically, because the excluded volume of a DNA bead is determined by the physical diameter of B-DNA (which is ~20 Å), and we use the repulsive part of a Lennard–Jones potential to model DNA–DNA repulsion (see below), the coarsest resolution that a DNA bead can have in our model is 5 bp (as this corresponds to ~17 Å in length). Such a resolution guarantees that the distance among sequentially bonded DNA beads is smaller than their excluded volume size, and thus, makes the DNA a self-avoiding polymer. A higher resolution would work too, but would introduce unnecessary computational expense, and would reduce the size of the chromatin systems we could study. The resolution of histone proteins, in contrast, can be reduced significantly more. We choose a resolution of one bead per histone octamer, as this is sufficient to represent the geometry of the nucleosome, the topological distribution of DNA beads around the histone core, and the chemically-specific strength of the nucleosome–nucleosome interactions. Together, the resolution of our minimal model allows us to reduce the number of particles in a 300 kb chromatin region, or 1500 nucleosomes, from ~30 million atoms (plus solvent) to only ~50,000 beads.

When devising our minimal model, we took care in capturing the manner in which nucleosomes interact with one another and the mechanical properties of the DNA (torsional and bending flexibility). Both features are crucial in defining the conformational space of nucleosomes. To account for these two features, while enabling efficient sampling of a few thousand nucleosomes, our minimal chromatin model approximates the shape and size of the histone core with a single bead (using an ellipsoid of 28 × 28 × 20 Å radii), and represents linker and nucleosomal DNA with one finite-size orientable sphere (using an ellipsoid of 12 × 12 × 12 Å radii) for every 5 base-pair steps. Inspired by the success of the RBP model in adequately approximating the atomistic mechanical properties of DNA at a single base-pair-step resolution, we propose a "minimal RBP-like model" for DNA at a resolution of 5 base pairs per bead. Accordingly, we optimize the minimal helical parameters from chemically-specific coarse-grained simulations of 200 bp DNA strands (Supplementary Methods). Then we describe nucleosome–nucleosome and nucleosome–DNA interactions by a series of orientationally dependent potentials fitted to reproduce internucleosome potentials of mean force calculated with our chemically-specific coarse-grained chromatin model (Supplementary Fig. 4 and

Supplementary Methods). To represent breathing nucleosomes, we analyze our chemically-specific chromatin simulations to determine the fraction of DNA that remains predominantly bound to the histone core and define only that fraction as nucleosomal DNA. For each snapshot in the chemically-specific model trajectories, we will get a different definition which builds up a set of structures representative of the thermodynamic equilibrium distribution of nucleosome breathing states. By using diverse structures from this set, we can incorporate the effects of nucleosome breathing in the model without having to directly simulate completely free DNA, significantly reducing the degrees of freedom, enabling superior sampling. In the case of the non-breathing nucleosomes, we simply define the nucleosomal DNA following the same definition used in the chemically-specific model. Then, in both cases, breathing and non-breathing nucleosomes, we attach permanently the nucleosomal DNA to the histone core bead. Hence breathing and non-breathing nucleosomes are constructed using equilibrium configurations from different chemically-specific simulations (Supplementary Fig. 3). In addition, to account for the slightly exposed histone core in breathing nucleosomes, we add an additional anisotropic potential to the total energy. We parameterize this anisotropic term to be consistent with experimental force spectroscopy experiments on single nucleosomes[114–117] (Supplementary Methods and Supplementary Figs. 2, 4). A list of all the parameters we use in this minimal chromatin model, along with the choice of values and justification is given in Supplementary Table 10.

**Debye-length Hamiltonian replica exchange**. To drive chromatin systems over the free-energy barriers and achieve complete sampling, we developed a Hamiltonian exchange method that attempts exchanges between replicas with different values of the Debye-length of the screened-coulomb interaction ($\lambda_D$) while keeping the temperatures the same. As in standard Hamiltonian exchange, the exchange probability is given by:

$$P(i \leftrightarrow i+1) = \min\left(1,\ \exp\left[\frac{1}{k_B T}\left(U_{\lambda_D^i}(\mathbf{x}^i) - U_{\lambda_D^i}(\mathbf{x}^{i+1}) + U_{\lambda_D^{i+1}}(\mathbf{x}^{i+1}) - U_{\lambda_D^{i+1}}(\mathbf{x}^i)\right)\right]\right), \quad (1)$$

where $\mathbf{x}^i$ are the chromatin coordinates of the $i$th replica, and $U_{\lambda_D^i}$ the potential energy function at Debye length $\lambda_D^i$—the original Debye length of the $i$th replica. The exchange is accepted or rejected based on the Metropolis criteria, and upon exchange, the potential energy functions (or coordinates) are switched. For our 12-nucleosome chromatin systems, we find that at the value of $\lambda_D = 8.0$ Å (corresponding to 0.15 mol/L salt), the chromatin structures are compact and suffer from sampling issues, increasing the Debye length to 15 Å gives rise to open structures that can sample effectively. We find that a range of 8.0–15.0 Å requires 16 replicas to get an acceptance probability of 30 %. This is significantly less than the ~80 replicas that we found would be required for standard T-REMD to sample the range of 300–600 K with a similar exchange probability. Additionally, the different Debye lengths are all at physically relevant salt concentrations and 300 K, therefore while increasing the sampling we can investigate the salt-dependent behavior of the system. Therefore, without the need for reweighing, the information extracted directly from each of the different replicas can be compared with experimental observables at different salt concentrations, whereas for T-REMD only the replica at 300 K (or the experimental temperature) is directly (ie., without reweighing) relevant for experimental comparison.

**Software used**. Simulations were performed using LAMMPS[154] (version 3rd March 2020) with our custom code (see Code availability). We used the program 3DNA[158] (version 2.3) within our model building methods. All data analysis was done using Python (version 3.8.5) with NumPy (version 1.19.2) and SciPy (version 1.5.2). All data was plotted using Matplotlib (version 3.3.2). Images were rendered using the Open Visualization Tool (OVITO) software[159] (version 3.0.0). We used the Weighted Histogram Analysis Method (WHAM) program[160,161] (version 2.0.9) to calculate PMFs.

**Reporting summary**. Further information on research design is available in the Nature Research Reporting Summary linked to this article.

## Data availability

Data supporting the findings of this manuscript are available from the corresponding author upon reasonable request. A reporting summary for this Article is available as a Supplementary Information file. The source data are available at https://doi.org/10.6084/m9.figshare.13663859.v1.

## Code availability

The authors are delighted to share the computational implementation of their models with the community. All the necessary files can be found at https://github.com/CollepardoLab/CollepardoLab_Chromatin_Model and https://doi.org/10.6084/m9.figshare.13663685.v1. Please use them freely and remember to cite this paper and LAMMPS (http://lammps.sandia.gov)[154]. The authors are happy to answer any questions and comments by email, and welcome contributions for any updates.

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

## Acknowledgements

This project has received funding from the European Research Council (ERC) under the European Union's Horizon 2020 research and innovation program (grant agreement No 803326). S.E.F. would like to acknowledge the EPSRC Centre for Doctoral Training in Computational Methods for Materials Science for funding under grant number EP/L015552/1. E.J.W. is funded by an EPSRC studentship from grant EP/R513180/1. A.G. is funded by an EPSRC studentship from grant EP/N509620/1. R.C.-G. is an Advanced Fellow from the Winton Programme for the Physics of Sustainability. J.A.J. is a Research Fellow at King's College. This work has been performed using resources provided by the Cambridge Tier-2 system operated by the University of Cambridge Research Computing Service (http://www.hpc.cam.ac.uk) funded by EPSRC Tier-2 capital grant EP/P020259/1.

## Author contributions

R.C.-G. and S.E.F. conceived the project; S.E.F., E.J.W., and R.C.-G. designed the models and the computational framework; S.E.F. performed research and analyzed data; A.G. contributed to the computational implementation of the model; J.A.J. contributed to the interpretation and contextualization of the data; R.C.-G. wrote the manuscript with help from S.E.F, J.A.J., and E.J.W. All authors reviewed the manuscript; R.C.-G. acquired funding; and R.C.-G. supervised research.

## Competing interests

The authors declare no competing interests.
