## [Peer Review File · Nature Communications]

Reviewer #1 (Remarks to the Author):

The authors develop a family of models creating a multi-scale approach to study chromatin structure and dynamics. Current experiments observe these effects at larger scales (tens of kb) or at the nucleosome level. Little is known during the transition region between these two scales, i.e., between 500 bases to a few kb, which is the typical size for genes. This is the region of interest in this paper! The authors comment that most of the co-polymer models at several kb resolution have been trained on experimental data and do not provide a direct quantitative connection to physical parameters. To achieve this goal, this paper utilizes, according to the authors, a “multiscale chromatin model that allows us to investigate the connection between the fine molecular details of nucleosomes (e.g., amino-acid and DNA sequence heterogeneity, specific distributions of post-translational modifications and epigenetic marks, protein flexibility, DNA mechanics, and nucleosome plasticity) and the mesoscale (up to sub-Mb scale) organization of chromatin”.

Although I like the approach proposed here, I have several concerns that preclude me to reach a final decision before I have some clarifications;

1. At the atomistic level, it is clear that electrostatic effects play a major role. Therefore, I am concerned if the current description for the “chemically-specific chromatin model” is sufficient to quantitatively observe the necessary plasticity. For example, recent studies in RNA (particularly riboswitches) require Mg to be treated as particles and not in a continuum fashion (See for example, Roy et al, NAR, 47, 3158, 2019). Therefore, I am not sure if this current approach is sufficient to quantitatively reproduce the observed plasticity of the nucleosome).
2. There is a need for a better justification for the three levels of coarse graining shown in Figure 1. There is very little explanation about the details of the choices for the coarse graining models. If all the authors wanted to do was to come a good qualitative description that would be sufficient. However, they criticize earlier models for not being able to directly connect to physical parameters – they should, therefore, quantitatively justify their choices. I do not believe that few comparisons to a selected choice of experiments is sufficient. A more robust justification is needed.
3. One of the more interesting results of the paper is the comparison of structural ensemble for breathing versus non-breathing nucleosomes. It makes a lot of sense and explains some of the liquid nature properties of the structural ensemble of chromatin (chromatin is a viscoelastic system). However, is this quantitatively correct? One needs a better justification.

If the authors are able to address all these concerns, the paper would be much stronger and then I would feel more comfortable making a final determination on its suitability to Nature Communications.

Reviewer #2 (Remarks to the Author):

What are the physical and molecular factors that modulate the accessibility of nucleosomes within compact assemblies? To resolve this conundrum, Farr et al. assessed how the mesoscale properties of chromatin were impacted by the dynamic behavior of nucleosomes using a computational modeling. The authors first developed a multiscale chromatin model to investigate the connection between the fine molecular details of nucleosomes and the mesoscale chromatin organization. They observed that the nucleosome breathing, which is enhanced at physiological salt conditions,

destabilized the formation of regular 30-nm fibers, and drove chromatin to a highly dynamic but compact, liquid-like organization. Interestingly, this nucleosome plasticity enhanced the multivalency of nucleosomes and, therefore, the connectivity and stability of both compact chromatin and the phase-separated condensed chromatin liquid. They suggested the stochastic organization of nucleosomes within compact chromatin contributes to the modulation of DNA accessibility. The research aim and obtained findings are exciting and of importance to cell biologists. I would support this paper. For publication in Nature Communications, my specific comments are as follows:

Major comments:

1) The authors checked the energetic consistency between level 2 and level 3 CG models. Then, the minimal CG model allowed for calculating a phase diagram regarding the mesoscopic nucleosome-nucleosome organization, which revealed a thermodynamically stable structure of the system. I wonder how long the time-scale was extended in the CG from level 2 to level 3 since, in general, CG makes the time resolution vague while I consider that this point must be an advantage of CG.

2) There seems to be no description of the linker histones in Methods and Results. It might be interesting to discuss how the linker histones can work in the authors' model.

3) Fig. S10. When the authors used 12-nucleosome arrays with 195 bp repeats, the results seem similar between the non-breathing and breathing conditions. It might be better to mention that the longer DNA linkers can intrinsically create the nucleosome–nucleosome orientation heterogeneity that sustains chromatin's liquid-like behavior.

Minor comments:

1) Page 2. Regarding "the complex and crowded biomolecular environment of the cell nucleus", density of the total materials in the chromatin environment was optically measured and reported (PMID: 28835378).

2) Fig. 1. For lay readers, note that "CG" is for "coarse-grained".

3) It was reported that binding of DNA-bending non-histone proteins also destabilizes the regular 30-nm fiber (PMID: 28135276).

Statement on the Revision of ⟨Manuscript ID NCOMMS-20-46992⟩ Based on the Referees' Reports

Stephen E. Farr, Esmee J. Woods, Jerelle A. Joseph, Adiran Garaizar,
Rosana Collepardo-Guevara

February 2, 2021

Comments by Reviewer #1

The authors develop a family of models creating a multi-scale approach to study chromatin structure and dynamics. Current experiments observe these effects at larger scales (tens of kb) or at the nucleosome level. Little is known during the transition region between these two scales, i.e., between 500 bases to a few kb, which is the typical size for genes. This is the region of interest in this paper! The authors comment that most of the co-polymer models at several kb resolution have been trained on experimental data and do not provide a direct quantitative connection to physical parameters. To achieve this goal, this paper utilizes, according to the authors, a "multiscale chromatin model that allows us to investigate the connection between the fine molecular details of nucleosomes (e.g., amino-acid and DNA sequence heterogeneity, specific distributions of post-translational modifications and epigenetic marks, protein flexibility, DNA mechanics, and nucleosome plasticity) and the mesoscale (up to sub-Mb scale) organization of chromatin". Although I like the approach proposed here, I have several concerns that preclude me to reach a final decision before I have some clarifications

Response: We thank the reviewer for their encouraging comments, we are delighted to hear that they liked our approach.

1. At the atomistic level, it is clear that electrostatic effects play a major role. Therefore, I am concerned if the current description for the "chemically-specific chromatin model" is sufficient to quantitatively observe the necessary plasticity. For example, recent studies in RNA (particularly riboswitches) require Mg to be treated as particles and not in a continuum fashion (See for example, Roy et al, NAR, 47, 3158, 2019). Therefore, I am not sure if this current approach is sufficient to quantitatively reproduce the observed plasticity of the nucleosome).

Response: We agree with the reviewer that considering the effects of Mg²⁺ when investigating DNA-histone interactions is important. Mg²⁺ is highly abundant inside cells, with the total concentrations estimated at ~10-20 mM and the free fraction at less than 5% of this (DOI: 10.1016/j.cub.2017.12.035). Low concentrations of Mg²⁺ ions are known to modestly increase the compaction of chromatin in vitro (DOI: 10.1038/emboj.2012.80). The idea is that Mg²⁺ ions outcompete Na⁺ ions (DOI: 10.7554/eLife.44993), and interact preferentially with the multiple closely-spaced negative charges on the nucleosome surface and on the DNA; hence, screening the overall DNA-DNA repulsion more effectively. However, regarding nucleosome unwrapping, reassuringly, the landmark work of the late Jonathan Widom (DOI: 10.1038/nsmb801) showed that in the presence of 100 mM Na⁺, addition of 0-10 mM of Mg²⁺ did not substantially change the equilibrium constant for nucleosome unwrapping. Consistently, more recent single-molecule FRET studies show that increasing the ionic strength of the buffer, by raising instead the NaCl concentration to up to 400 mM, has a minimal impact on nucleosome unwrapping (DOI: 10.1038/s41467-018-06758-1). Therefore, the effects that increased screening by Mg²⁺ ions at physiological concentrations could exert in the modulation of nucleosome unwrapping are likely subtle and take place at much finer scales than what we set to capture with our chemically-specific model. Importantly, the experimental data that we have used to quantitatively validate the force-induced unwrapping response of our model (i.e., mechanical unzipping experiments of Wang and colleagues)

is for nucleosomes already immersed in a buffer containing both 100 mM Na⁺ and 0.5 mM Mg²⁺, demonstrating that the balance of potentials and parameters in our model captures well this regime.

In general, accounting for the specific effects of polyvalent ions in protein and nucleic acid behaviour with molecular modelling is far from trivial. For such a task, high-resolution biomolecular models—typically atomistic but at the very least one-bead per heavy atom descriptions—plus explicit ions are a must. However, even full atomistic descriptions of biomolecules with explicit water and ions using empirical force fields are known to unphysically overbind polyvalent ions to proteins and nucleic acids (DOI: 10.1063/5.0017775). These issues become particularly relevant when trying to predict the fine atomistic structures or detailed folding pathways of RNA motifs and non-canonical DNAs (e.g., quadruplexes), as these can be highly sensitive to, and in many cases only stable upon, specific binding of polyvalent ions. Indeed, the study suggested by the reviewer shows that Mg²⁺ ions crucially shift the balance between the ON/OFF atomistic structures adopted by the SAM-I riboswitch; the key in such a study is the combination of atomistic models for the riboswitch, an explicit model for Mg²⁺ ions, and experimental data (which, crucially, is used to compensate for the missing physical details of the models). However, unlike riboswitches, chromatin is far too large to be sampled effectively at such a level of detail. Furthermore, even if we decided to reduce the scope of our work and focus on much smaller systems—integrating an explicit description of ions within a model of nucleosomes at high-resolution—this would involve many approximations too, and would not automatically increase the accuracy of the predictions we could make.

In our model, all the atomic charges, pseudo-charges, dipoles, and quadrupoles within a single base pair or a single amino acid are clumped together into a single particle. Consistently with this approximation, we describe electrostatic interactions simply by means of the Debye-Hückel approximation, which is a mean field theory where the ion density is given by the linearized Boltzmann distribution, and where effects like ion correlations, ion heterogeneity and specific ion binding are ignored. By doing this, we assume that the effect of counterions is simply to screen the mean electrostatic potential from the chromatin charges, and that charge-charge interactions decay like a Yukawa function with distance. Although approximate, such a treatment is crucial to reduce the high dimensionality of the chromatin phase-space, which in turn is what allows us to achieve complete sampling with our Debye-length sampling method and obtain an energy landscape for force-induced unwrapping of nucleosomes in quantitative agreement with that derived from experiments.

We have now added a discussion to the main manuscript acknowledging the limitations of our description of counterions in solution.

2. There is a need for a better justification for the three levels of coarse graining shown in Figure 1. There is very little explanation about the details of the choices for the coarse graining models. If all the authors wanted to do was to come a good qualitative description that would be sufficient. However, they criticize earlier models for not being able to direct connect to physical parameters - they should, therefore, quantitatively justify their choices. I do not believe that few comparisons to a selected choice of experiments is sufficient. A more robust justification is needed.

Response: We thank the reviewer for identifying a point that needed clarification. Accordingly, we have extensively revised the description of our models in the main manuscript. Additionally, we want to clarify that our intention was never to criticize data-driven models. We think both mechanistic and data-driven models offer complementary, and equally valuable, information on chromatin organization. We have now tried to emphasize this in the manuscript. Indeed, we believe that to achieve the “holy grail” of resolving whole nucleus chromatin structure with sub-nucleosome details, one of the most promising strategies would be to integrate a mechanistic model like ours, to a polymer model derived from whole nucleus experimental data sets.

In addition to the updated description of the models in the main manuscript, we have also added two new Supporting Tables (Table IX and Table X) that provide a summary of all the respective model parameters, their values, and a justification of the different choices. Below, we summarize the rationale we used when choosing the different model resolutions.

Level 1: At the highest level of resolution (level 1), we use atomistic models because such resolution is needed to elucidate the sequence-dependent properties of DNA and the secondary structural preferences and flexibility of proteins.

Level 2: When designing our chemical-specific coarse grained model (level 2), our goal was to fulfill two opposing requirements: (1) maintaining a high-enough resolution to capture with molecular accuracy the effects of DNA and protein sequence and breathing motions of nucleosomes, and (2) reducing the number of degrees of freedom as much as possible to achieve simulations of oligonucleosome systems. For this, we considered that a key feature of nucleosomes, for chromatin compaction, is that they have a charged and contoured surface and also 10 highly positively charged and flexible histone tails that extend from their surface. While histone core proteins are largely α -helical and exhibit relatively small structural fluctuations vs. those of histone tails in crystallographic studies

and in atomistic MD simulations, histone tails are largely disordered and flexible. To capture these features, we needed to describe histone proteins at the residue level; this is the coarsest resolution that still allow us to preserve the chemical composition of this histone proteins in full, model the secondary structure of the histone core proteins and secondary structural changes (if needed in future), describe the flexibility of histone tails and its modulation by post-translational modifications, incorporate chemical modifications (in progress), interrogate the role of specific histone-residue mediated interactions to chromatin organization, probe binding of additional proteins to specific chromatin residues (in progress), and describe the contributions of distinct amino acids to nucleosome unwrapping. Similarly, when designing the DNA part of our chemically specific model, we chose one bead per basepair, as that is the coarsest resolution that still allowed us to capture in full the influence of sequence in the mechanical properties of DNA (i.e., twist, roll, and twist). The resolution of the combined model allows us to reduce the number of atoms in, for example, a 5 kb chromatin region, or ~ 25 nucleosomes, from ~ 0.5 million (plus solvent) to only $\sim 25,000$ beads.

Level 3: In a similar spirit as the design of our chemically-specific model, our aim with the minimal model (level 3) was to find the coarsest possible representation of nucleosomes that would simultaneously (1) allow us to reproduce the chromatin structural ensembles that we observe with our chemically-specific model, and (2) increase our capacity to model much larger chromatin systems (i.e., with thousands of nucleosomes and reach functionally-relevant scales or phase-separating systems). To accurately represent the difference between the breathing and non-breathing behaviour of chromatin that we observe in the chemically specific simulations, and the torsional and bending properties of the linker DNA (which we found determine the configurational ensembles of chromatin), we require an explicit representation of the DNA—this is indeed one of the limiting factors defining the model resolution. The specific resolution of the DNA model is dictated by the dimensions of the DNA. That is, because the excluded volume of a bead is fixed by the diameter of DNA (which is approximately 20 \AA), the coarsest resolution that a DNA bead can have in our model is 5 bp (which corresponds to approximately 17 \AA in length), as this guarantees that the excluded volume size of the DNA beads is sufficient to prevent DNA polymers from unphysically passing through each other. A higher resolution would work too, but would introduce unnecessary computational expense, reducing the size of the chromatin systems we could study. The resolution of histone proteins is chosen as 1 bead per one histone octamer, as this is sufficient to represent the geometry of DNA binding to the histone core. Furthermore, we find that the simple combination of an attractive core bead with repulsive DNA beads is sufficient to capture the orientation dependent inter-nucleosome interactions, this approximates the fact that DNA is strongly negatively charged while the histone core has net positive charge. Together, the resolution of our minimal model allows us to reduce the number of atoms in, for example, a 300 kb chromatin region, or ~ 1500 nucleosomes, from ~ 30 million (plus solvent) to only $\sim 50,000$ beads.

3. One of the more interesting results of the paper is the comparison of structural ensemble for breathing versus non-breathing nucleosomes. It makes a lot of sense and explains some of the liquid nature properties of the structural ensemble of chromatin (chromatin is a viscoelastic system). However, is this quantitatively correct? One needs a better justification.

Response: We thank the reviewer for their encouraging comments and identifying a point that needed further clarification. We are very pleased that they found our structural ensembles for breathing versus non-breathing interesting and capable of explaining some of the liquid nature of chromatin. While some of the predictions of our model are in quantitative agreement with experiments (as we describe next), some of the trends that we describe are only qualitative. We have now clarified this throughout our manuscript.

As we mention above, the quantitative agreement includes the energy landscape for nucleosome unwrapping at single base pair resolution in our model versus that computed from the single-molecule unzipping experiments of Wang and colleagues. For 12-nucleosome 165-bp chromatin, our sedimentation coefficients are also in quantitative agreement with the experimental values of Grigoriev and colleagues for the exact same system and conditions. In addition, our chemically-specific chromatin model is based on a high-resolution x-ray nucleosome structure, and preserves the fine-molecular topology of the nucleosome. The charge, masses, size, and hydrophobic properties of amino acids and DNA bases, as well as the flexibility of histone tails and sequence-dependent DNA strands are derived from the atomistic structure and atomistic simulations of nucleosomes and DNA, which were validated also by comparing quantitatively against experimental data.

Our model qualitatively recapitulates the progressive inhibition of nucleosome breathing as salt decreases from 0.15 M to 0.01 M observed experimentally. The disordered chromatin ensembles that we report are in qualitative agreement with those observed by various important experiments, such as the nucleosomes clutches model derived from super-resolution microscopy of chromatin in nuclei and the polymorphic structure of chromatin revealed by chromEMT. The importance of nucleosome plasticity for chromatin LLPS is also in qualitative agreement with the recent groundbreaking experiments on HP1-mediated chromatin LLPS of Serena Sanulli and colleagues. However, as any other molecular model, ours is still an approximate description of chromatin. We do not know of any quantitative experimental data that characterizes the full structural ensembles that chromatin populates, so although we know

that our observations align with a wide-range of observations on the liquid-like character of chromatin, unfortunately, we do not know if our predictions are quantitative or not. Nevertheless, the ability of our models to reveal how the intrinsic plasticity of nucleosomes—a fundamental, yet largely ignored, property in mechanistic coarse-grained models—underpins the propensity of chromatin to form highly compact but stochastic arrangements, represents an important advancement in our mechanistic understanding of chromatin organisation. We have now added a statement to clarify that our results are expected to be, at worst, in qualitative agreement with experiments.

If the authors are able to address all these concerns, the paper would be much stronger and then I would feel more comfortable making a final determination on its suitability to Nature Communications.

Comments by Reviewer #2

What are the physical and molecular factors that modulate the accessibility of nucleosomes within compact assemblies? To resolve this conundrum, Farr et al. assessed how the mesoscale properties of chromatin were impacted by the dynamic behavior of nucleosomes using a computational modeling. The authors first developed a multiscale chromatin model to investigate the connection between the fine molecular details of nucleosomes and the mesoscale chromatin organization. They observed that the nucleosome breathing, which is enhanced at physiological salt conditions, destabilized the formation of regular 30-nm fibers, and drove chromatin to a highly dynamic but compact, liquid-like organization. Interestingly, this nucleosome plasticity enhanced the multivalency of nucleosomes and, therefore, the connectivity and stability of both compact chromatin and the phase-separated condensed chromatin liquid. They suggested the stochastic organization of nucleosomes within compact chromatin contributes to the modulation of DNA accessibility. The research aim and obtained findings are exciting and of importance to cell biologists. I would support this paper. For publication in Nature Communications, my specific comments are as follows:

Major comments: 1) The authors checked the energetic consistency between level 2 and level 3 CG models. Then, the minimal CG model allowed for calculating a phase diagram regarding the mesoscopic nucleosome-nucleosome organization, which revealed a thermodynamically stable structure of the system. I wonder how long the time-scale was extended in the CG from level 2 to level 3 since, in general, CG makes the time resolution vague while I consider that this point must be an advantage of CG.

Response: We thank the reviewer for this interesting suggestion. Motivated by this, we have performed additional simulations to measure the rates of exponential decay of the autocorrelation function of the radius of gyration of a 12-nucleosome chromatin system with both of our coarse-grained models (i.e., level 2 and level 3). We have included these analyses in a new figure in the SI (Figure S5), and have added this information in the ‘Multiscale model for heterogeneous chromatin’ section of our paper and in more detail to the Supporting Information. We find that the correlation time for our minimal model (level 3) is ten times smaller than the correlation time for our chemically-specific model (level 2), which indicates that the timescales in level 3 are a factor of ten faster than those in level 2.

2) There seems to be no description of the linker histones in Methods and Results. It might be interesting to discuss how the linker histones can work in the authors’ model.

Response: Although our approach can consider linker histones and other proteins of interest that bind to chromatin, we have intentionally left linker histones out of this work, as we are currently conducting a follow up study that focuses on their effects. However, motivated by the reviewer comment, and to emphasize the ability of our technique to consider binding of additional proteins to chromatin, we have now added a description of this in Methods. In short, linker histones, and other proteins of interest, can be described in the exact same way as the core histone proteins. Using a high resolution structure of the protein (e.g. DOI: 10.1016/j.molcel.2015.06.025 for H1/H5) as the reference and either structural information or atomistic simulations, we would first define the protein regions that are likely to remain disordered or adopt a stable secondary fold (taken from the high-resolution data). With this information, we would build a one bead per amino acid model, where each bead would carry the charge, size,

mass and hydrophobic character of the amino acid it represents. Finally, beads within globular domains would be connected by an elastic network model, and those within disordered regions would be described as fully flexible chains.

3) Fig. S10. When the authors used 12-nucleosome arrays with 195 bp repeats, the results seem similar between the non-breathing and breathing conditions. It might be better to mention that the longer DNA linkers can intrinsically create the nucleosome-nucleosome orientation heterogeneity that sustains chromatin's liquid-like behavior.

Response: We thank the reviewer for this excellent suggestion, we have now rephrased this in the main text as suggested.

Minor comments: 1) Page 2. Regarding "the complex and crowded biomolecular environment of the cell nucleus", density of the total materials in the chromatin environment was optically measured and reported (PMID: 28835378). 2) Fig. 1. For lay readers, note that "CG" is for "coarse-grained". 3) It was reported that binding of DNA-bending non-histone proteins also destabilizes the regular 30-nm fiber (PMID: 28135276).

Response: We have addressed all the minor comments as well.

Reviewer #1 (Remarks to the Author):

I am happy with the improvement of the manuscript with the new changes by the author and I believe it is ready for publication now.

Reviewer #2 (Remarks to the Author):

The authors addressed all of my comments adequately. The revised manuscript was much improved. I would support the publication of this paper.